# Histone H4 lysine 16 acetylation controls central carbon metabolism and diet-induced obesity in mice

Cecilia Pessoa Rodrigues[1,2,3,6], Aindrila Chatterjee[1,4,6], Meike Wiese [1], Thomas Stehle[1], Witold Szymanski [5], Maria Shvedunova [1] & Asifa Akhtar [1,2 ✉]

Noncommunicable diseases (NCDs) account for over 70% of deaths world-wide. Previous work has linked NCDs such as type 2 diabetes (T2D) to disruption of chromatin regulators. However, the exact molecular origins of these chronic conditions remain elusive. Here, we identify the H4 lysine 16 acetyltransferase MOF as a critical regulator of central carbon metabolism. High-throughput metabolomics unveil a systemic amino acid and carbohydrate imbalance in *Mof* deficient mice, manifesting in T2D predisposition. Oral glucose tolerance testing (OGTT) reveals defects in glucose assimilation and insulin secretion in these animals. Furthermore, *Mof* deficient mice are resistant to diet-induced fat gain due to defects in glucose uptake in adipose tissue. MOF-mediated H4K16ac deposition controls expression of the master regulator of glucose metabolism, *Pparg* and the entire downstream transcriptional network. Glucose uptake and lipid storage can be reconstituted in MOF-depleted adipocytes in vitro by ectopic *Glut4* expression, PPARγ agonist thiazolidinedione (TZD) treatment or SIRT1 inhibition. Hence, chronic imbalance in H4K16ac promotes a destabilisation of metabolism triggering the development of a metabolic disorder, and its maintenance provides an unprecedented regulatory epigenetic mechanism controlling diet-induced obesity.

[1] Department of Chromatin Regulation, Max Planck Institute of Immunobiology and Epigenetics, 79108 Freiburg, Germany. [2] Faculty of Biology, University of Freiburg, Schaenzlestrasse 1, 79104 Freiburg, Germany. [3] International Max Planck Research School for Molecular and Cellular Biology (IMPRS-MCB), Freiburg, Germany. [4] European Molecular Biology Laboratory, Meyerhofstrasse 1, 69117 Heidelberg, Germany. [5] Proteomics Facility, Max Planck Institute of Immunobiology and Epigenetics, Freiburg, Germany. [6] These authors contributed equally: Cecilia Pessoa Rodrigues, Aindrila Chatterjee. ✉email: akhtar@ie-freiburg.mpg.de

Chronic diseases, in particular noncommunicable diseases (NCDs), are major health challenges for modern societies and account for over 70% of deaths worldwide. Metabolic disorders such as obesity and hyperglycemia are considered major risk factors for both development and maintenance of NCDs[1,2].

Epigenetic regulators have been implicated in molecular regulation of metabolic pathways and are thereby responsible for keeping the metabolic state of the organism in check[3–5]. Central players of major metabolic reaction modules are transcriptionally[6,7] and post translationally[8] regulated by epigenetic regulators. For example, transcription factors such as PGC1α[9], C/EBPα[10], or CRTC2[11] responsible for control of metabolic networks are dynamically regulated by acetylation in response to metabolic stimuli. Consequently, alterations in levels of epigenetic regulators can translate into metabolite imbalances. In fact, both histone methyltransferase and histone deacetylase enzymes have been implicated in the regulation of obesity, including as factors mediating resistance to diet-induced obesity[12–15]. However, the contribution of a large and important class of chromatin-modifying enzymes, the lysine acetyltransferases (KAT), to these processes has not been systematically described to date.

MOF (also known as KAT8) is the major KAT responsible for acetylation of histone H4 at lysine 16 (H4K16ac), a unique histone mark that directly modulates higher-order chromatin structure[16] and the only acetylation mark known to be inter-generationally maintained[17]. Genome-wide association studies have identified MOF as a risk locus for multiple traits frequently associated with obesity and T2D, including waist circumference[18], serum triglyceride (TG) levels[19] and body mass index (BMI)[20–23]. Whether changes in MOF levels are causative of these phenotypes as well as the nature of the underlying pathogenic mechanisms remain enigmatic.

In this study, we systematically interrogate the metabolic roles of MOF across the body using metabolite profiling of six metabolically important organs. Complete MOF knockout is deleterious for cells and tissues[24–28] and causes embryonic lethality in mice[29]. However, Mof haploinsufficiency is tolerated in mice[30] and humans[31]. Thus, we used Mof heterozygous mice ($Mof^{+/-}$) to dissect the metabolic consequences of reduced MOF activity in multiple tissues. Metabolomic analyses revealed that reduction of this central KAT results in global remodeling of primary nutrient metabolism, including altered carbohydrate and amino acid metabolism in multiple tissues. At the physiological level, these mice show disrupted glucose assimilation and impaired insulin response, suggestive of a predisposition to developing a metabolic syndrome. Indeed, when challenged with a high-fat diet (HFD), $Mof^{+/-}$ animals exhibit impaired fat storage and resistance to weight gain. We demonstrate that MOF-mediated H4K16ac is required for the transcriptional regulation of PPARγ and the downstream transcriptional network mediating glucose uptake and neutral lipid storage in white adipose tissue.

Taken together, we reveal MOF and its major target H4K16ac as the first KAT involved in diet-induced obesity resistance. Furthermore, we implicate MOF in regulation of systemic glucose and amino acid balance and suggest that reduced MOF levels predispose animals to a metabolic syndrome with features common to T2D.

## Results

### Chronic Mof depletion is associated with systemic deregulation of amino acid and carbohydrate metabolism.
For an organism-wide overview, we generated metabolomic profiles of seven functionally diverse tissues: heart, brain, spleen, kidney, skeletal muscle (SKM), liver, and serum from 8-week-old $Mof^{+/-}$ mice and littermate controls (Fig. 1a, Supplementary Fig. 1a–c). We identified 1,365 unique features across all tissues. The datasets clustered according to their tissue of origin in sparse partial least squares discriminant analysis (sPLS-DA), indicating tissue-specific differences (Fig. 1b). This was mirrored by their ionMz features abundance profiles and associated pathways (Fig. 1c, Supplementary Fig. 1d–e). A survey of significantly deregulated features (unique ionMz, $q \le 0.1$) also revealed that different tissues mounted distinct metabolite responses to altered Mof levels (Fig. 1d). Whilst the spleen ($n = 380$), brain ($n = 281$), heart ($n = 209$) and serum ($n = 293$) showed significant alterations, impacts on the liver ($n = 22$), SKM ($n = 36$) and kidney ($n = 4$) were much milder (Fig. 1d, Supplementary Data 1).

Given that every organ has a unique set of metabolic requirements, we expected that altered Mof dosage would produce disparate effects on individual tissues. In spite of this, differentially expressed metabolites (DMs) in spleen, heart, and brain (the most strongly affected tissues) belonged predominantly to three major primary nutrient classes: amino acids, lipids, and carbohydrates as well as their derivatives (Fig. 1e, f, Supplementary Fig. 1d). In $Mof^{+/-}$ brain samples, the strongest effect was scored in the central carbon metabolism pathways, including deregulation of glycolysis and pyruvate metabolism (Fig. 1f, Supplementary Fig. 1d, e). The brain and heart, organs which are characterized by a high energetic expenditure, both showed an increase in metabolites related to energy metabolism (Fig. 1f). The heart and spleen showed marked reductions in multiple features that were annotated as proteinogenic amino acids and associated pathways (Fig. 1f). All tissues showed deregulation of multiple amino acid metabolism pathways (valine, leucine/isoleucine, glycine, serine threonine, alanine, aspartate, and glutamate), the non-oxidative branch of the pentose phosphate pathway, TCA, as well as vitamin B5 and B6 metabolism. Despite their inherent physiological differences, multiple organs show recurring patterns of deregulation in similar metabolic pathways following MOF reduction, serving as crucial indicators of systemic metabolite destablization in Mof heterozygous animals.

Next, we focused on DMs in the serum. Its connection to all organ systems make the serum valuable for obtaining a broad overview of circulating metabolites and identifying potential systemic effects. We scored a significant correlation between metabolites found in the serum and those found in the heart, SKM, and spleen (Supplementary Fig. 1g). Detailed pathway enrichment analysis of serum DMs revealed two important trends. First, most of the significantly deregulated pathways are also deregulated in at least one other tissue set (Fig. 1f, g). Second, metabolism of alanine, aspartate, glutamate, and pentose phosphates, which are consistently deregulated in the heart, spleen, and brain, also show disruption in the serum (Fig. 1g highlighted in red). Central carbon and amino acid metabolism were the most significantly altered classes of metabolite in the serum of $Mof^{+/-}$ mice (Fig. 1g).

Chronic metabolic rewiring as observed in $Mof^{+/-}$ mice can result in pathologies. For example, an altered serum glutamine to glutamate ratio has emerged as a reliable biomarker for predicting metabolic disorders such as T2D[32]. The circulating serum metabolite profile of $Mof^{+/-}$ mice showed depletion of a number of features matching the mass of glucogenic amino acids (glutamic acid, aspartic acid, and glycine), elevated glutamine and a concomitant increase in hexose (glucose) levels (Fig. 1g, h; Supplementary Data 1 and 2). This is indicative of a possible shift in organ nutrient utilization, probably triggered by impaired glucose consumption. Despite elevated levels of circulating hexose in the serum, we detected reduced hexose and increased lactate in $Mof^{+/-}$ brains, the highest glucose-consuming organ of the body. We also observed increased consumption of alternative fuel sources including acetate, primary amino acids (and derivatives)

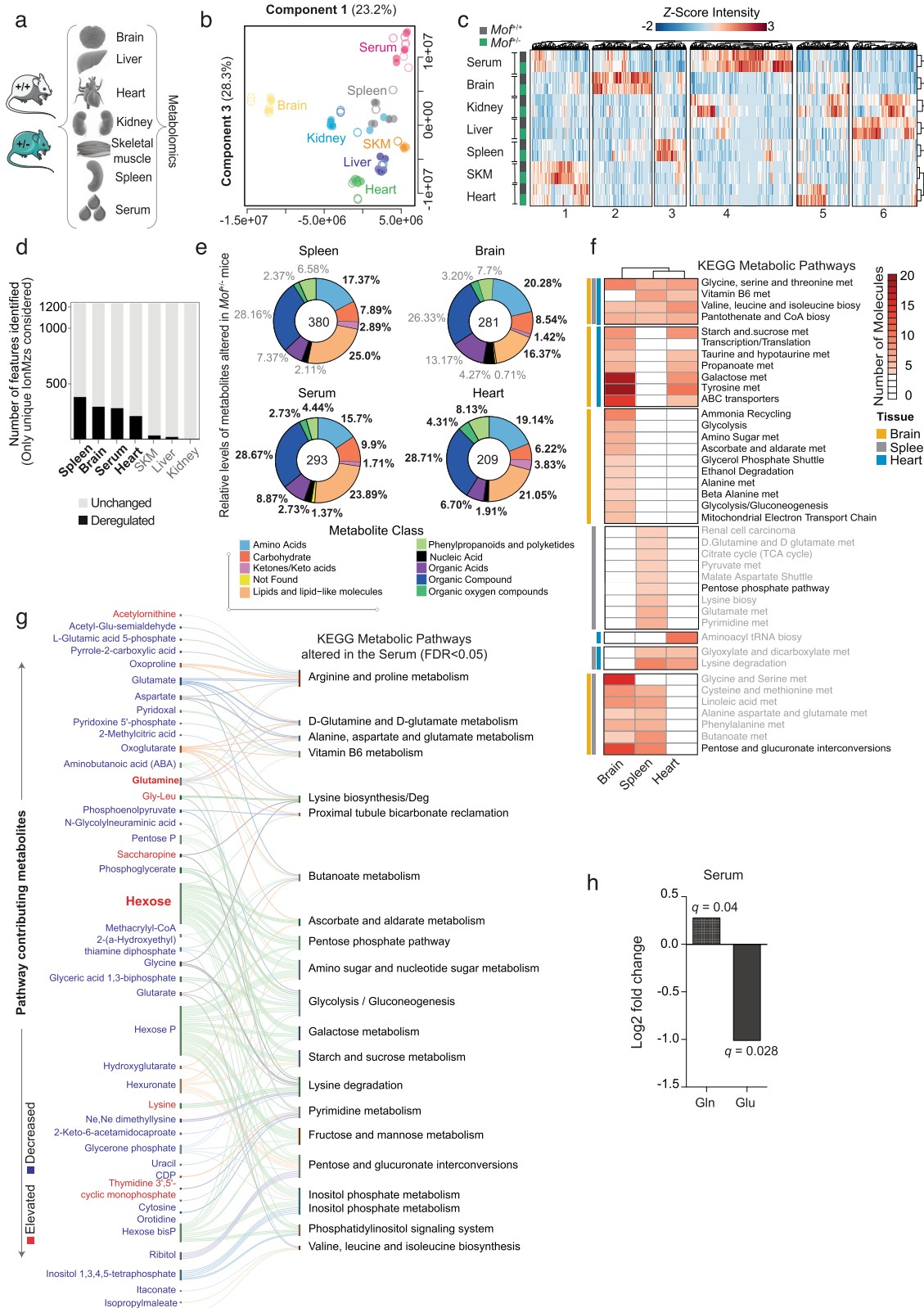

as well as ketone derivatives in $Mof^{+/-}$ hearts. Ketone bodies are generated in the liver for extrahepatic consumption[33], providing an important source of nutrition under starvation and particularly in diabetic states[34]. Notably, livers of $Mof^{+/-}$ mice showed an accumulation of ketone derivatives (Supplementary Fig. 1d–f), suggesting that energy metabolism is disrupted in $Mof^{+/-}$ animals.

Given a potential global deregulation of energy metabolism in $Mof$ heterozygous animals, we decided to complement our analysis by concentrating on the visceral white adipocyte tissue (WAT) as the major energy reservoir of the body. To this end, we performed polar and apolar metabolomics from WAT (Supplementary Fig. 2a). Similar to our observations in brain, heart, and spleen, we detected an impairment of carbohydrate and amino

**Fig. 1 Chronic reduction of MOF triggers systemic metabolic destablization. a** Experimental overview of metabolite extraction scheme from different tissues of 8-week-old wild-type (+/+, gray, $n = 4$) and *Mof* heterozygous (+/−, green, $n = 3$) mice. *See also:* Supplementary Fig. 1a–c. **b** Partial least squares discriminant analysis (PLS-DA) scores plot of identified features across all seven analyzed tissue types. Two components covering maximum feature variance are shown. $Mof^{+/+}$-Hollow circles; $Mof^{+/−}$-Filled circles. **c** Heatmap representing relative levels of different metabolites across all six tissues and the serum of both genotypes. Unsupervised *k*-means clustering is used to segregate the data into six major classes, closely mirroring tissue-specific metabolite signatures. *See also:* Supplementary Fig. 1d. **d** Stacked-plot showing the differentially regulated (black) and total number of metabolites (gray) found per tissue. Bold black labels represent tissues with significant enrichment $p < 0.05$ scored by two-sided Fisher's exact test. **e** Donut-plot showing the distribution of metabolite classes from significantly deregulated features, based on the highest probability annotations. Key metabolite classes are highlighted as bold. *See also:* Supplementary Fig. 1e–g. **f** Heatmap of the enriched deregulated pathways found in the brain (yellow), spleen (gray), and heart (blue). Color-scale depicts the number of contributing DMs in the pathway. KEGG pathway names denoted in black are common to all three tissues. **g** Sankey-plot showing deregulated pathways in serum and contributing metabolites (based on most probable annotation from deregulated feature ionMz). Metabolite colors depict direction of change, up- (red) and down- (blue) regulation ($q$ value < 0.1). **h** Glutamine and glutamate Log$_2$ fold change found in serum of $Mof^{+/−}$ animals. *Post hoc* calculation of $q$-value was performed using the Storey method[97] and significance set as $q$ value < 0.1 (details in "Methods"). *See also:* Supplementary Fig. 1 and 2; Supplementary Data 1 and 2.

acid metabolism in WAT of $Mof^{+/−}$ mice (Supplementary Fig. 2b, c). Remarkably, we found that $Mof^{+/−}$ adipocytes showed significant deregulation of the insulin pathway (Supplementary Fig. 2b), suggesting that $Mof^{+/−}$ animals may be more susceptible to glucose deregulation.

Our systematic metabolomic profiling unveils a role for MOF in controlling the levels of the major nutritional sources in multiple organs. Collectively, the overarching deregulated pathways in the sampled tissues were related to central carbon and amino acid metabolism. Amino acid levels can impact insulin secretion and are frequently associated with metabolic syndromes[35–39]. Thereby, we hypothesize that *Mof* haploinsufficiency may affect insulin sensing or signaling, leading to aberrant global glucose homeostasis and thereby predisposing $Mof^{+/−}$ animals to the development of metabolic disorders.

**Mof haploinsufficient mice show impaired glucose sensing.** To investigate the physiological consequences of the systemic central carbon metabolism imbalance, we metabolically phenotyped young (8-week-old) and old (26-week-old) $Mof^{+/−}$ animals. We detected no significant differences in body mass or composition between $Mof^{+/−}$ and $Mof^{+/+}$ animals at either age (Fig. 2a). Oral glucose tolerance testing (OGTT) in young $Mof^{+/−}$ mice revealed impaired glucose tolerance, which was further aggravated in older animals (Fig. 2b). $Mof^{+/−}$ mice showed delayed glucose absorption and a significantly higher glycemic peak 15 min after oral glucose administration (Fig. 2c). Furthermore, overnight-fasted $Mof^{+/−}$ mice showed attenuated insulin production upon glucose challenge compared to littermate controls, despite secreting similar levels of insulin at steady state (Fig. 2d-e). We conclude that $Mof^{+/−}$ animals show significant defects in glucose sensing and impaired glucose-stimulated insulin secretion.

We could recapitulate these glucose sensing defects in vitro using differentiated endocrine cells derived from the human PANC-1 cell line transfected with *MOF* siRNAs (Supplementary Fig. 3a, b). MOF-depleted endocrine cells show attenuated insulin production and reduced insulin mRNA expression upon glucose challenge (Supplementary Fig. 3a). Hence, MOF levels likely influence insulin production and/or secretion upon glucose challenge.

Intriguingly, we observed a strong upregulation of insulin receptor substrate 1 (*Irs1*) mRNA in the pancreas, heart, and SKM of $Mof^{+/−}$ mice (Supplementary Fig. 3c), which may reflect a compensatory mechanism to cope with impaired glucose sensing. Importantly, no change in the frequency of insulin-producing cells (Supplementary Fig. 3d), nor in the size, morphology, endocrine organization or inflammation status of the pancreatic islets was observed in $Mof^{+/−}$ mice

(Supplementary Fig. 3e). Collectively, these data indicate that *Mof* reduction affects glucose-sensing rather than production. MOF is a transcriptional regulator[26,40]. To investigate whether the glucose sensing defect observed after *Mof* reduction was underpinned by specific gene expression changes, we took advantage of publicly available scRNA-seq datasets from different tissues of healthy wild-type mice[41] (Fig. 2f). These data were used to generate a gene regulatory network (GRN)[42] setting *Mof* as the regulatory gene node. This analysis allowed us to predict candidate genes that are regulated by *Mof* and subsequently classify them according to their metabolic pathways. This unbiased analysis identified enrichment of pathways related to insulin signaling and T2D predisposition in multiple organs, thereby predicting a correlation between *Mof* levels and diabetes predisposition on the gene regulatory level (Fig. 2f). We furthermore correlated the fold changes (FC) of DM detected in the liver and SKM of $Mof^{+/−}$ mice with those detected in the same tissues in prediabetic and diabetic human patients[37,43]. Indeed, we observed a positive correlation between the metabolites upregulated in $Mof^{+/−}$ mice and increasing during the progression from prediabetic to diabetic state in human patients[37] (Fig. 2g), with amino acid metabolism representing the strongest deregulated class in both species (Supplementary Fig. 3f).

Collectively, these data identify defective glucose assimilation and impaired insulin secretion as the major pathological consequences of *Mof* haploinsufficiency. We hypothesize that $Mof^{+/−}$ animals are susceptible to development of a metabolic syndrome that has the potential to evolve into T2D.

**Mof haploinsufficient mice are resistant to HFD-induced obesity.** To test if reduction of MOF can indeed increase the risk of developing a metabolic syndrome, we challenged $Mof^{+/−}$ mice with nutrient stress. A key trigger for metabolic disorders is high caloric food intake. We therefore fed 7-week-old $Mof^{+/−}$ and $Mof^{+/+}$ mice a HFD *ad libitum*. Strikingly, after 26 weeks of HFD, $Mof^{+/−}$ mice gained ~50% less body weight compared to controls (Fig. 3a, b). This surprising resistance to weight gain in HFD was predominantly caused by the inability of $Mof^{+/−}$ mice to increase fat mass, while lean mass, food intake, total water content, and basal temperature were unaffected (Supplementary Fig. 4a–c). Resistance to HFD-induced weight gain of $Mof^{+/−}$ mice showed no sex bias, with a similar phenotype observed in both male (Fig. 3a) and female (Supplementary Fig. 4d, e) animals. We also found that *MOF* mRNA levels positively correlate with the body mass index (BMI) of healthy human subjects (Fig. 3c), suggesting that MOF-mediated regulation of weight gain may be conserved in mammals.

To infer the immediate response of $Mof^{+/−}$ mice to HFD, we focused on earlier time points. After 1 week of HFD challenge, $Mof^{+/−}$ mice showed an elevated glycemic peak 15 min post oral

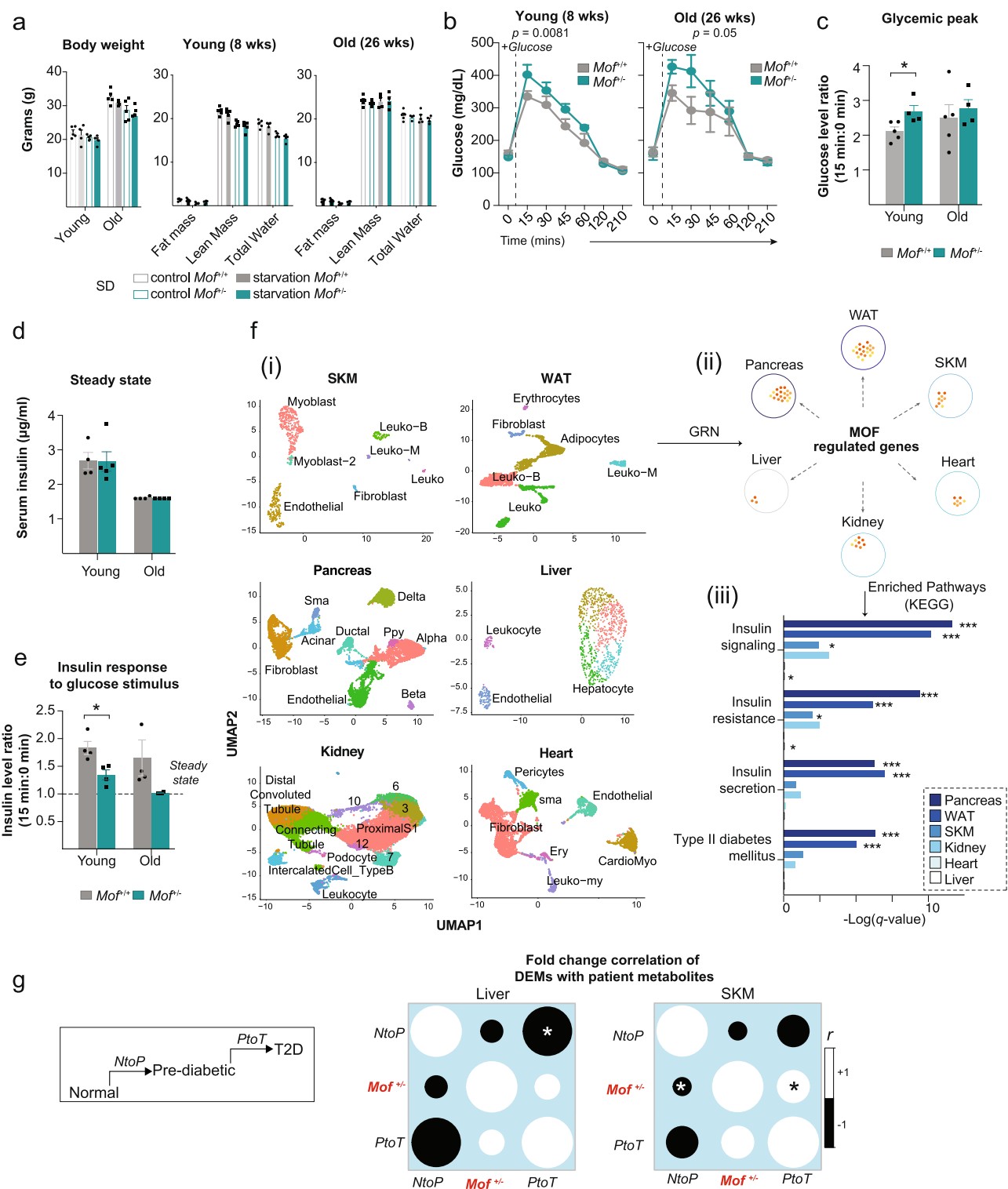

gavage (Fig. 3d, e, Supplementary Fig. 4f), similar to their response under standard diet (SD) conditions (Fig. 2), as well as reduced insulin secretion (Fig. 3f). After 8 weeks of HFD, control animals showed elevated basal glucose levels and a higher OGTT glycemic peak (Fig. 3d, Supplementary Fig. 4f), accompanied by increased body weight. At the same time, $Mof^{+/-}$ mice showed higher glycaemia, but a sharp resistance to gaining weight or fat mass (Fig. 3b, Supplementary Fig. 4b, d, e). A long-term consequence of persistent elevation in serum glucose levels is the modification of hemoglobin to glycated hemoglobin (HbA1c).

We compared HbA1c levels (significance threshold of 6%[44,45]) in the sera of $Mof^{+/-}$ mice fed SD or HFD (Fig. 3g). After 26 weeks of SD, $Mof^{+/-}$ mice already showed >6% HbA1c levels. Continued HFD exposure for 26 weeks resulted in elevated serum HbA1c levels (>6%) of control animals, while $Mof^{+/-}$ mice showed a more pronounced response of >10% HbA1c levels. We observed that the free water content of $Mof^{+/-}$ mice on SD was comparable to that observed in the (obese) $Mof^{+/+}$ mice on HFD, therefore not only corroborating the hypothesis that reduced $Mof$ levels are associated with metabolic syndrome predisposition, but

**Fig. 2 _Mof_ haploinsufficient mice show impaired glucose tolerance and insulin secretion. a** Barplots showing the total body weight (left), overall percentage of lean and fat mass, and total water of 8-week-old ("young", middle) or 26-week-old ("old", right) $Mof^{+/+}$ and $Mof^{+/-}$ animals under steady state (open bars) or after 16 h of starvation (filled bars). Dots in the barplots represent independent animals and error bars represent ±SEM. $n = 4$–5/ group. **b** Blood glucose levels before and after oral glucose tolerance test (OGTT); 1 g/kg mouse; $n = 5$/group in 8- (young, left) and 26-week-old (old, right) $n = 5/Mof^{+/+}$ and $n = 4/Mof^{+/-}$ male mice. Statistical analysis performed by two side, two-way ANOVA. $p$ values are indicated in the figure. Error bars represent means ± SEM. **c** Glycemic peaks are represented as a ratio of peak glucose level (15 min after oral administration) over steady-state (fasting) level. Statistical analysis performed by two-sided Mann–Whitney test, *$p = 0.05$. Error bars indicate ±SEM. $n = 4$–5/group. Serum insulin levels at (**d**) steady state (fasting) and (**e**) at maxima (15 min after oral glucose administration), represented relative to fasting levels. Statistical analysis performed by two-sided Mann–Whitney test, *$p = 0.05$. Biological replicates: young: $Mof^{+/+}$ $n = 5$; $Mof^{+/-}$ $n = 4$ and old: $Mof^{+/+}$ $n = 5$ and $Mof^{+/-}$ $n = 4$. Error bars indicate ±SEM. **f** (i) UMAP-plots showing the major populations found in single-cell data from wild-type skeletal muscle (SKM), pancreas, white adipocytes (WAT), liver, kidney, and heart. Data were obtained in the PangloDB dataset (top panel). (ii) scRNA-seq data from indicated tissues was used to infer a gene regulatory network (GRN) using _Mof_ as the regulatory node. Edges represent gene subsets. (iii) KEGG Gene Set Enrichment Analysis results depicted in barplots (colored by tissue type). Benjamini–Hochberg method applied to calculate FDR adjusted $p$ values: ***$p = 0.001$, and *$p = 0.05$. See also Supplementary Source file for the original scRNA-seq data[41,114]. **g** Comparative analysis of the tendencies of identified mouse metabolites that were previously scored in transition from normal to prediabetic (NtoP) and subsequently to diabetic stages (PtoT) in patients; original data from[37]. Color bar indicates Spearman correlation scores ($r$); *$p < 0.05$. See also: Supplementary Fig. 3.

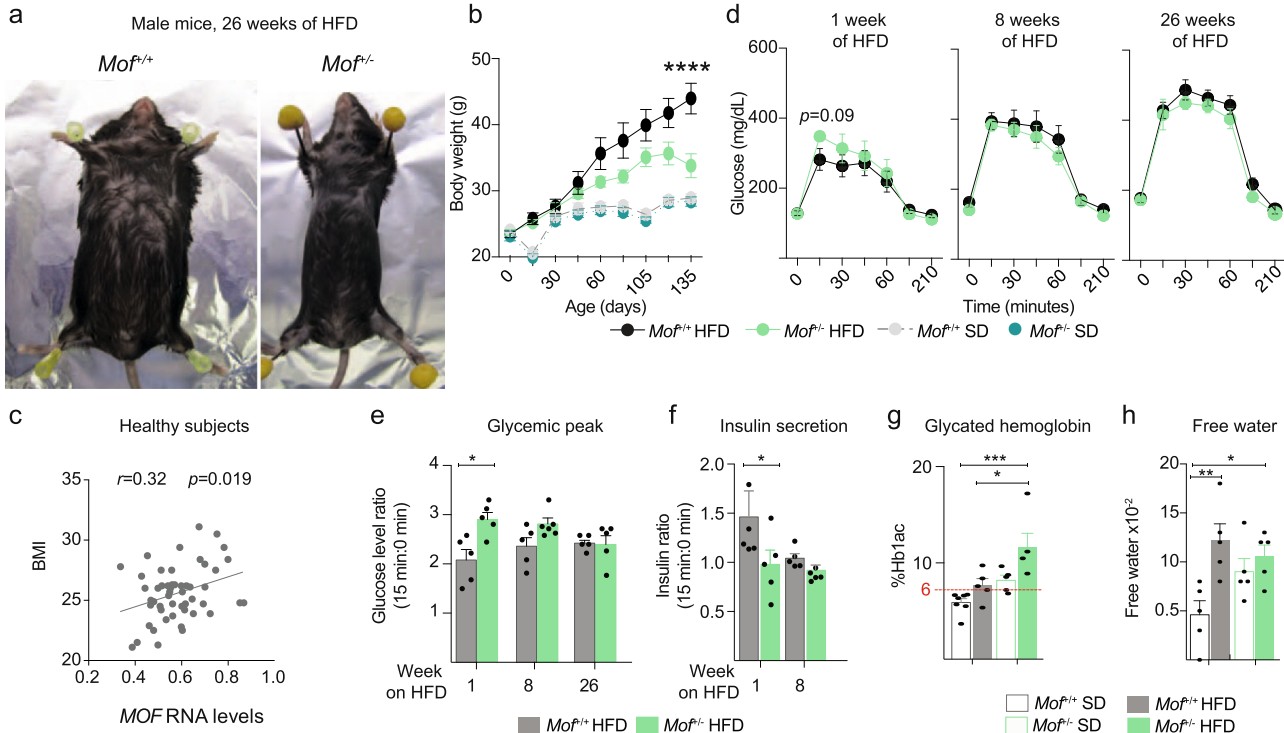

**Fig. 3 _Mof_ heterozygous mice are resistant to HFD-induced fat mass gain. a** Representative images of male control and $Mof^{+/-}$ mice after 26 weeks of HFD. See also Supplementary Fig. 4a–d for female. **b** Body weight gain under standard (SD) and high-fat diets (HFD). Statistical analysis performed by two-way ANOVA, ****$p = 10^{-16}$. SD $Mof^{+/+}$ $n = 5$, SD $Mof^{+/-}$ $n = 4$, HFD $Mof^{+/+}$ $n = 5$, HFD $Mof^{+/-}$ $n = 5$. Error bars indicate ±SEM. See also: Supplementary Fig. 4d. **c** Overall correlation between _MOF_ trimmed mean of M (TMM) values normalized mRNA[115] levels and body mass index (BMI). Correlation score and $p$ value calculated by Pearson. Original data from GSE50244. **d** Blood glucose levels before and after OGTT (Glucose bolus 1 g/kg mouse). Assay performed after 1, 8, and 26 weeks of HFD treatment in $Mof^{+/+}$ (black) and $Mof^{+/-}$ (green) mice ($n = 5$/group). Error bars indicate ±SEM. Statistical analysis performed by one-way ANOVA, followed by Tukey test. Significant ($p < 0.05$) $p$ value is indicated in the figure. See also: Supplementary Fig. 4f. **e** Glycemic peak, represented as a ratio of peak glucose level (15 min after oral administration) relative to steady-state (fasting) level. Error bars indicate ±SEM. Statistical analysis performed by one-way ANOVA, followed by Tukey test *$p = 0.04$. $n = 5$/group. **f** Peak serum insulin levels (15 min after oral glucose administration) relative to steady state (fasting) levels. Error bars indicate ±SEM. Statistical analysis performed by one-way ANOVA, followed by Tukey test *$p = 0.02$ $n = 5$/group. **g** Percentage of glycated hemoglobin (HbA1c) in SD ($Mof^{+/+}$ $n = 7$; $Mof^{+/-}$ $n = 5$) or HFD ($n = 5$/group). Error bars indicate ±SEM. Statistical analysis performed by one-way ANOVA followed by Tukey test, *$p = 0.02$, ***$p = 0.0004$. **h** Free water in SD or HFD as a percentage of total body composition ($n = 5$/group). Statistical analysis performed as in (**g**). Statistical analysis performed by two-sided one-way ANOVA followed by Tukey test, *$p = 0.04$, **$p = 0.006$. Error bars indicate ±SEM. See also: Supplementary Fig. 4.

also ruling out the possibility that weight loss in $Mof^{+/-}$ animals is caused by water deprivation (Fig. 3h).

The phenotypes exhibited by $Mof^{+/-}$ mice following the HFD challenge provide strong evidence for a metabolic predisposition of these animals toward a prediabetic state. To substantiate this hypothesis, we specifically searched for further evidence of multi-organ comorbidities that are frequently associated with a prediabetic state[46–48] (Supplementary Fig. 4g–m). HFD-challenged $Mof^{+/-}$ mice showed a significant decrease in pancreatic islet area compared to control animals (Supplementary

Fig. 4g). HFD treatment resulted in increased inflammation in spleens and livers of both $Mof^{+/+}$ and $Mof^{+/-}$ mice when compared to their respective SD controls (Supplementary Fig. 4h–j). However, $Mof^{+/-}$ mice showed a more severe inflammatory response in both organs (Supplementary Fig. 4h–j).

Disruption in the balance between pro-inflammatory T helper 17 cells (Th17) and anti-inflammatory regulatory T cells (Treg) is a well-known hallmark of chronic tissue inflammation[49–51]. Chronic metabolic disorders, such as diabetes, promote the expansion and polarization of Th17 cells[52–54], and interference with this process has been shown to mitigate diabetic symptoms in patients[55]. Spleens of both control and $Mof^{+/-}$ animals displayed increased frequencies of Th17 cells and concomitant decreases in the Treg cell population under HFD conditions (Supplementary Fig. 4k). In stark contrast, livers of HFD-fed $Mof^{+/-}$ animals displayed a significantly higher frequency of pro-inflammatory Th17 cells than control animals (Supplementary Fig. 4l, m).

Notably, the liver is one of the main organs showing increased frequency of intra-tissue Th17 cells during diabetic onset[56,57]. To validate the tissue-intrinsic contribution of MOF to the aggravated inflammatory response in the liver microenvironment, we generated liver organoids from HFD-challenged $Mof^{+/-}$ mice and performed co-culture experiments with wild-type naïve T helper cells (CD4+ cells) (Supplementary Fig. 4n, o). We scored increased polarization of activated T cells toward the Th17 profile specifically in cells that were co-cultured with liver organoids derived from HFD-fed $Mof^{+/-}$ mice (Supplementary Fig. 4o). Our results suggest that the unique liver microenvironment of $Mof^{+/-}$ animals can polarize T cells toward a pro-inflammatory Th17 profile upon HFD exposure.

Taken together, $Mof^{+/-}$ animals show a striking resistance to HFD-induced gains in fat mass. The resistance to diet-induced obesity is accompanied by a marked deregulation of glucose homeostasis, elevated glycated-hemoglobin levels, and multi-tissue inflammation which combined are strong evidence for a diabetic predisposition caused by $Mof$ haploinsufficiency.

**MOF regulates the gene expression profile of adipose tissue.** In order to understand the molecular mechanisms underlying the resistance to HFD-induced obesity, we performed mRNA-seq of visceral WAT, which are the main fat reserves in adult mice (Fig. 4a, Supplementary Data 3). First, we compared the gene expression profiles from SD-fed 8-week-old $Mof^{+/+}$ and $Mof^{+/-}$ mice to identify the differences at baseline prior to nutrient stress (Fig. 4a, DEG$_{geno}$). Although neither genotype displayed gross phenotypic changes under SD conditions, the PCA analysis could already discriminate a clear separation between the gene expression profiles of $Mof^{+/+}$ and $Mof^{+/-}$ mice (Supplementary Fig. 5a). $Mof^{+/-}$ WAT showed significant downregulation of genes associated with inflammation, tissue remodeling, and angiogenesis (Supplementary Fig. 5b–d)—all pathways associated with the development of obesity.

Next, to take both the impact of genotype and food intake into account, we cross-compared differentially regulated genes ($p < 0.05$) identified upon SD (Supplementary Fig. 5b) with those identified in mice fed a HFD for 26 weeks (Supplementary Fig. 5c). We performed unbiased clustering (based on expression similarity) on the differentially regulated genes in all four datasets combined (4,088 genes). Out of five identified clusters, cluster 1 (593 genes) showed the highest expression in $Mof^{+/-}$ mice independently of their diet and is therefore the most insightful for uncovering MOF-specific functions in WAT (Fig. 4b). Interestingly, these genes are associated with metabolic pathways such as endocrine disorders, glucagon signaling, parathyroid hormone, and insulin resistance (Fig. 4b, Supplementary Fig. 5d and

Supplementary Data 3). Other clusters of this analysis showed gene expression profiles characteristic of responses to HFD treatment (clusters 4 and 5) and the development of obesity (cluster 3) (Supplementary Fig. 5d, Supplementary Data 3).

To identify the pathways under direct regulation by MOF, we performed MOF ChIP-seq in WAT isolated from young $Mof^{+/+}$ and $Mof^{+/-}$ animals fed a SD. We identified 19,753 peaks (mapping to 9,423 genes) in $Mof^{+/+}$ WAT and 716 peaks (mapping to 444 genes) in $Mof^{+/-}$ WAT (Supplementary Data 3 and 4). In both datasets, MOF peaks were enriched at promoters (±1 kb). Control peaks can be grouped into two main clusters ($kmeans = 2$): cluster 1 (regions = 8,254 mapping to 4,439 genes) showing a mild decrease; and cluster 2 (regions = 11,486 mapping to 6,570 genes) showing a very clear reduction in the $Mof^{+/-}$ dataset (Fig. 4c, Supplementary Fig. 5e). Overlapping the MOF-bound genes of both clusters with significantly downregulated genes identified in the $Mof^{+/-}$ WAT (SD) RNA-seq (Supplementary Fig. 5b) revealed that the vast majority (81.5%) of downregulated genes are direct targets of MOF (Fig. 4d). Notably, pathway enrichment analysis revealed a strong association with central carbon metabolism and inositol phosphate metabolism/signaling (Fig. 4d). In WAT, phosphoinositides are not only important signaling molecules, but are key modulators of energetic balance in response to metabolic stress[58–60]. Hence, our genome-wide analysis uncovers a mechanistic link between MOF-mediated gene regulation and the energetic demand for fat storage in adipose tissues. Besides the prominent metabolic signature, many direct MOF targets in WAT were also related to tissue inflammation (Fig. 4d)—an important driver for obesity which appeared to be lost in the adipose tissue of $Mof^{+/-}$ animals.

A large proportion of MOF targets is regulated by deposition of H4K16ac. We cross-compared the list of genes downregulated in $Mof^{+/-}$ WAT (SD) against existing H4K16ac ChIP-seq profiles[61] and found a significant enrichment of H4K16ac domains at these genes (Supplementary Fig. 5f), suggesting they are likely to be regulated by MOF via H4K16ac. Furthermore, we compared the list of downregulated genes with those scored in obese $Trim28^{+/D9}$ haploinsufficient animals[15]. This analysis revealed a robust anti-correlation (Supplementary Fig. 5g), implying that the metabolic disorder triggered by $Mof$ haploinsufficiency is distinct from the one found in $Trim28^{+/D9}$ obese mice and therefore unique to MOF function.

To gain mechanistic insight into how $Mof^{+/-}$ animals react to HFD stress, we compared the transcriptional changes elicited by HFD treatment between $Mof^{+/+}$ and $Mof^{+/-}$ animals (Fig. 4a, Supplementary Fig. 5h, DEG$_{diet}$). In wild-type animals, HFD treatment results in activation of pathways associated with the development of obesity, such as carbohydrate metabolism, hormone and cytokine production, adipocyte differentiation, and lipid storage (Fig. 4e). Although $Mof^{+/-}$ mice showed hyper-activation of insulin signaling (Fig. 4b, qPCR validation in Supplementary Fig. 6d) and upregulation of lipid biosynthesis genes, genes involved in lipolysis or fatty acid uptake remained unaffected (Supplementary Fig. 6d). In agreement with this, our metabolomic analysis revealed deregulation of the insulin path-way in WAT (Supplementary Fig. 2b). We conclude that impaired lipid storage observed in WAT of HFD-induced obesity-resistant $Mof^{+/-}$ animals is neither caused by a defect in lipogenesis arising from impaired insulin signaling nor as a result of increased lipid breakdown or impaired lipid transport. In remarkable agreement with a role for MOF in regulating genes linked to energy balance and adipose tissue inflammation, these pathways are completely absent in HFD-treated $Mof^{+/-}$ animals (Supplementary Figs. 2 and 6).

To conclude, the transcriptome of $Mof^{+/-}$ WAT shows a downregulation of pathways associated with the development of

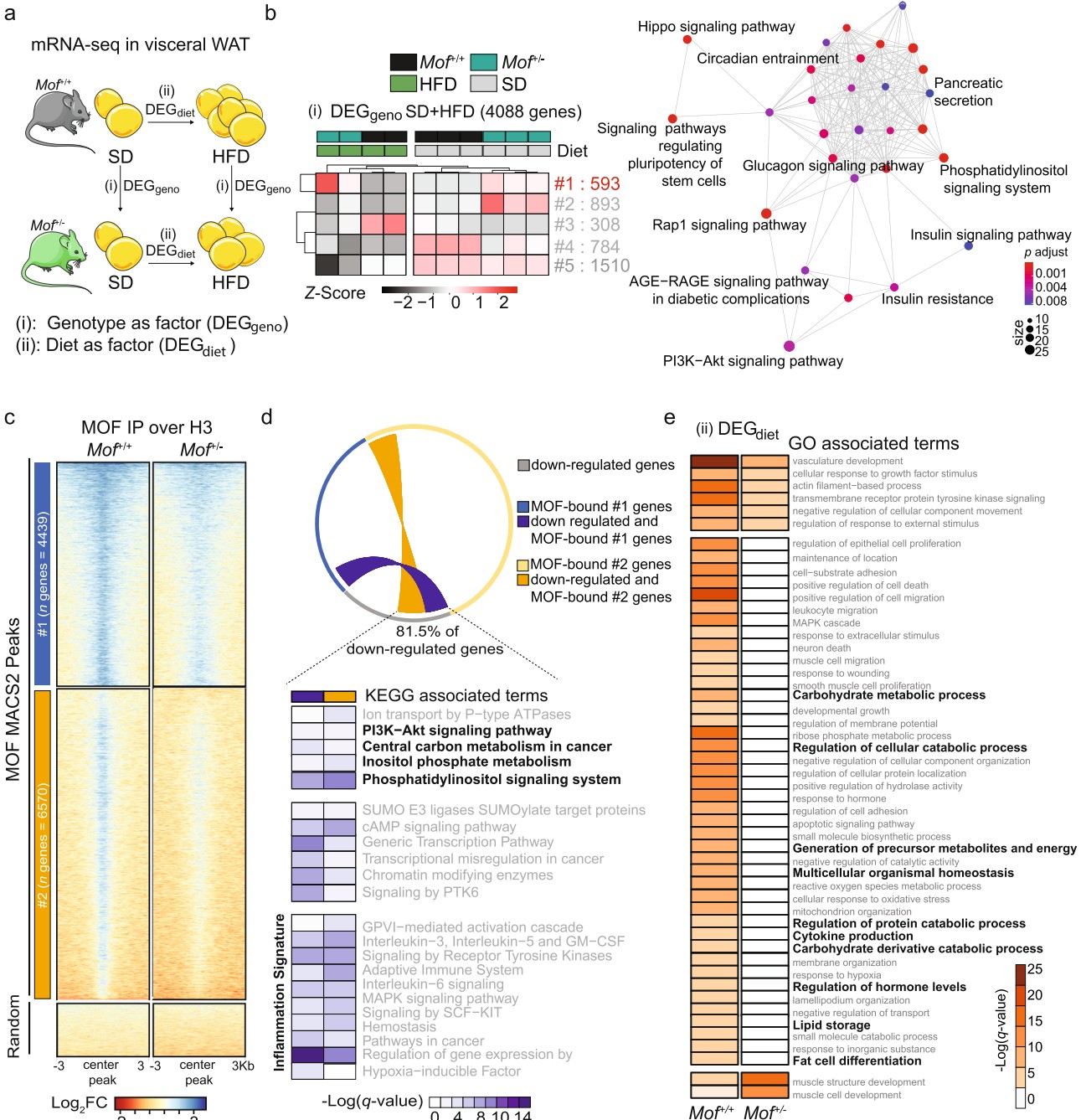

obesity. ChIP-seq analysis in the same tissue further identifies direct MOF/H4K16ac-mediated gene regulation of metabolic pathways associated with carbon metabolism. Taken together, molecular analyses indicate a critical role for MOF in balancing energy expenditure and storage in adipose tissue.

**Mof depletion results in impaired lipid storage and glucose uptake in adipocytes.** Our genome-wide analyses prompted us to examine the morphology of WAT after HFD treatment. Consistent with the transcriptional deregulation of lipid storage and other obesity-related pathways (Fig. 4e), we found the WAT of HFD-treated $Mof^{+/-}$ animals to be devoid of neutral lipid droplets (Fig. 5a, b, Supplementary Fig. 6a). Neither the overall number of mature adipocytes (Fig. 5c) nor the number of adipocyte progenitor cells (Supplementary Fig. 6b) were significantly decreased, suggesting that impaired lipid storage and lipogenesis,

rather than defective adipogenesis, are the predominant causes for obesity resistance in $Mof^{+/-}$ mice.

To narrow down the molecular mechanism for lipid storage defects observed in $Mof^{+/-}$ animals, we queried whether the genes in five selected gene ontology (GO) terms broadly related to adipocyte lipid metabolism were deregulated in WAT of $Mof^{+/-}$ animals under SD conditions. Among the selected GO terms, the only pathway significantly enriched for deregulated genes in $Mof^{+/-}$ WAT was glucose homeostasis (Fig. 5d and Supplementary Fig. 6c). This result was of particular significance as it closely supports our data showing MOF's involvement in regulating glucose homeostasis in vivo (Fig. 2a, b) and central carbon metabolism at the cellular level (Fig. 4d).

SKM and adipose tissue are the two major tissues responsible for absorbing glucose upon food intake, thereby regulating glycemia. In WAT, dietary sugars are converted into fatty acids

**Fig. 4 MOF-mediated transcription signature in adipose tissue. a** Schematic representation of strategy used for bulk mRNA-seq analysis of visceral WAT of $Mof^{+/+}$ and $Mof^{+/−}$ mice fed SD or HFD. Analysis (i) queries the differentially regulated genes (DEGs) between genotypes ($DEG_{geno}$) to determine effect of $Mof$ depletion. Analysis (ii) queries DEGs between diets ($DEG_{diet}$) independently for each genotype to determine effect of HFD. **b** Heatmap showing Z-score expression of $DEG_{geno}$ for SD and HFD treatments. $DEG_{geno}$ of SD and HFD were combined ($n = 4,088$) and unbiasedly grouped into five clusters based on expression similarity. A number of genes per cluster (right), genotypes (top; $Mof^{+/+}$: black and $Mof^{+/−}$: dark green), and diets (top; SD: gray and HFD: light green) are indicated. Network map (bottom) showing the significant enriched pathways for cluster 1 genes. Circles indicate the number of genes within each pathway. Color scale represents $p$ adjust values. See Supplementary Fig. 5a–d and Supplementary Data 3. False discovery rates (FDR) were controlled by the Benjamini–Hochberg method having $q$ value defined as 0.05. **c** Heatmaps showing the H3-normalized ChIP enrichment on MOF-bound peaks sorted by enrichment intensity and clusters based on unbiased clustering ($k$-means = 2). Enrichment signal is plotted relative to MOF-peak center ±3 kb. Color scale represents the $Log_2$ fold change. See also: Supplementary Fig. 5e and Supplementary Data 4. **d** Circos plot showing MOF ChIP-seq and mRNA-seq data integration. Significantly downregulated ($p < 0.05$) genes in $Mof^{+/−}$ are shown in gray, MOF-bound genes present in cluster 1 (#1, blue) or 2 (#2, yellow) are depicted. The inner lines show the common downregulated and bound genes in #1 (dark blue) or #2 (dark yellow). The lower panel displays the significantly enriched KEGG pathways associated with common genes. The scale shows the $−Log(q$ value). Pathways directly related to adipocyte energy metabolism are highlighted in bold. False discovery rates (FDR) were controlled by the Benjamini–Hochberg method having $q$ value defined as 0.05. **e** Significantly enriched pathways associated with changes in diet of $Mof^{+/+}$ and $Mof^{+/−}$ mice ($DEG_{Diet}$). Segments in the heatmap are separated (top to bottom) according to common deregulated pathways, pathways enriched only in $Mof^{+/+}$ and pathways enriched only in $Mof^{+/−}$. Scale shows $−Log(q$ value). Pathways related to the development of obesity are highlighted in bold. See also: Supplementary Fig. 5.

and subsequently stored as TGs. The insulin-sensitive glucose transporter GLUT4 is responsible for 70% of glucose uptake in adipocytes and SKM[62,63]. Interestingly, we detected a significant decrease of $Slc2a4$ (also known as $Glut4$) levels in WAT and SKM of HFD-fed $Mof^{+/−}$ animals (Fig. 5e, Supplementary Fig. 6d), suggesting possible perturbations of glucose import in these organs.

Alterations in GLUT4 levels may have dramatic effects on adipocytes' ability to store fat. In fact, transgenic mice over-expressing GLUT4 showed adipocyte hyperplasia due to increased lipogenesis[64–66]. Conversely, $Glut4$-null mice show strongly reduced adipose tissue deposits[67]. Accordingly, we observe a significant increase in levels of $Rbp4$ RNA (Supplementary Fig. 6c), encoding a transporter protein known to be upregulated in $Glut4$-null animals and contributing to insulin resistance in obese and T2D patients[68–70].

Since impaired lipid storage in WAT can be a direct outcome of defective adipocyte differentiation, we sorted mesenchymal stromal cells (MSCs) from $Mof^{+/+}$ and $Mof^{+/−}$ animals to monitor their differentiation in vitro (Fig. 5f, g). While $Mof^{+/+}$ MSCs showed normal differentiation toward the adipocyte lineage, they failed to increase their lipid reserves after glucose and insulin challenge (Fig. 5g). Thus, we could confirm that $Mof^{+/−}$ mice are able to maintain normal adipocyte differentiation, but the adipocytes fail to respond to glucose.

To investigate tissue-intrinsic effects of $Mof$ deletion on glucose flux in adipocytes, we took advantage of an in vitro conditional $Mof$ knockout system based on $Mof^{flox/flox}$ CreERT2$^{tg}$ (Cag-Cre/$Mof$-iKO) mice[26]. We isolated mouse embryonic fibroblasts (MEFs) from these mice and differentiated them into adipocyte-like cells (iAdipo)[71] (Fig. 5h, Supplementary Fig. 6e, f, see "Methods"). Upon 4-hydroxy-tamoxifen (4-OHT)-induced $Mof$ deletion (Supplementary Fig. 6g), $Mof$-iKO adipocytes showed decreased $Glut4$ mRNA and GLUT4 protein levels (Fig. 5i and Supplementary Fig. 6h) and hyper-activation of insulin signaling, fully recapitulating our observations from adipocytes in vivo (Fig. 4b, Supplementary Fig. 6i). Notably, along with low $Glut4$/GLUT4 levels $Mof$-iKO adipocytes showed impaired insulin-mediated glucose uptake (Fig. 5j). In addition, $Mof$-iKO adipocytes showed glucose-fueled elevation in extracellular acidification rate (ECAR), while glycolytic flux (Fig. 5k) and oxygen consumption rate (Supplementary Fig. 6j) were significantly reduced (Fig. 5k) in $Mof$-iKO iAdipo. This difference in glycolytic rates was even more pronounced after insulin-stimulated glucose uptake in these cells. While control cells

showed insulin sensitivity and a concomitant elevation in glycolytic rate, this response was abrogated in $Mof$-iKO adipocytes (Fig. 5k). Our in vitro analyses therefore support the hypothesis that $Mof$ reduction results in intracellular glucose deprivation in adipocytes.

$Mof^{+/−}$ animals show dramatic defects in lipid storage (Figs. 3a and 5a). Whilst lipid biosynthesis and transport genes are unaffected, we detect deregulation of glucose homeostasis genes. Moreover, $Mof^{+/−}$ animals show a downregulation of $Glut4$ in WAT, resulting in impaired glucose uptake.

**MOF regulates fat storage by regulating the $Glut4$ transcription network.** Glucose uptake is a requirement for neutral lipid storage in adipose tissue. To understand the interplay between defects in lipid storage and glucose uptake in $Mof^{+/−}$ animals, we challenged in vitro differentiated adipocytes (generated as in Fig. 5h) with glucose and insulin and performed neutral lipid staining. Similar to our observations in vivo, $Mof$-iKO adipocytes failed to increase their lipid storage upon combined glucose and insulin challenge (Fig. 6a, Supplementary Fig. 7a). Treating wild-type cells with MG149, an inhibitor with efficacy against MYST family HATs, including MOF[72], mimics the fat storage (Fig. 6a, Supplementary Fig. 7a) and glucose uptake defects (Supplementary Fig. 7d) in association with decreased levels of H4K16ac (Supplementary Fig. 7b, c). The loss of $Mof$ results in global reduction of H4K16ac in adipocytes in vitro (Supplementary Fig. 6g). We therefore treated $Mof$-iKO cells with Ex-527, an inhibitor of the HDAC SIRT1 responsible for deacetylation of H4K16ac[73–75]. Remarkably, Ex-527 treatment not only rescued the H4K16ac levels (Supplementary Fig. 7e), but also restored lipid storage in $Mof$-iKO cells (Fig. 6b, Supplementary Fig. 7a), indicating that chromatin regulation of H4K16ac is indeed essential. Notably, Ex-527 treatment significantly boosted lipid storage even in control cells. Moreover, we found that control iAdipo showed a significant increase in H4K16ac levels after glucose and insulin treatment, while $Mof^{+/−}$ cells failed to do so (Fig. 6c).

Lipid storage defects upon deletion of MOF could be rescued by ectopic expression of either $Mof$ or $Glut4$, while the expression of a $Mof$ catalytic mutant (E350Q) was unable to restore H4K16ac levels nor lipid storage defects (Fig. 6d, e, Supplementary Fig. 7a, f–h). These results imply that the impaired capacity of $Mof$-deficient adipocytes to store fat is, at least in part, due to defective glucose uptake by GLUT4. Treating $Mof$-iKO adipocytes with chloroquine—a drug that enhances GLUT4 translocation and fusion with the plasma membrane in response to insulin[76]—failed

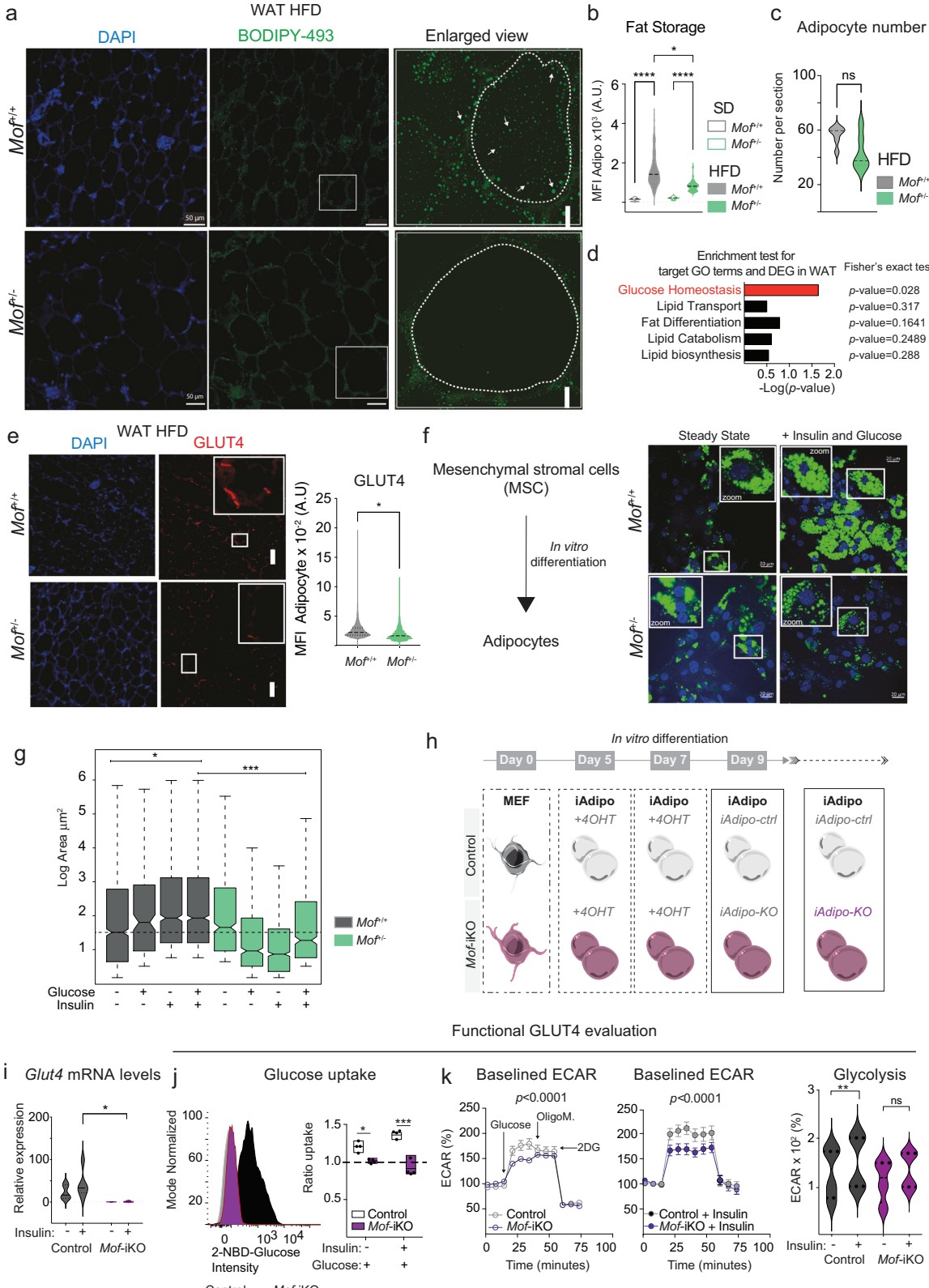

to restore fat storage in adipocytes (Fig. 6f, Supplementary Fig. 7a). This finding establishes that a decrease in GLUT4 levels, rather than defects in its intracellular transport, is the primary cause for reduced storage of neutral lipids in *Mof*-iKO adipocytes.

Given that *Mof* depletion results in downregulation of *Glut4* (Figs. 5e, 6g, Supplementary Fig. 6d), we next investigated

whether the transcriptional network regulating *Glut4* expression was affected in $Mof^{+/-}$ animals. We observed significant downregulation of the *Glut4* transcriptional activators *Pparg*, *Ppargc1a* (also known as: *Pgc1α*) and *Mef2c*[77,78], as well as of *Glut4* itself in WAT and SKM of SD-fed $Mof^{+/-}$ mice (Fig. 6g, Supplementary Fig. 7i–k). This pointed toward a perturbation of

**Fig. 5 *Mof* depletion impairs glucose uptake and fat storage in adipocytes. a** Immunofluorescence images of WAT after 26 weeks on HFD (SD controls are shown in Supplementary Fig. 6a). DAPI (blue) and neutral lipid staining by BODIPY-493 (green). Highlighted regions show recidual fat storage. Scale bar: 50 μm, enlarged view: 5 μm. **b** Violin-plot showing the MFI quantification of (**a**). Error bars indicate ±SEM. Statistical analysis was performed by one-way ANOVA followed by Kruskal–Wallis test, *$p = 0.04$, ****$p = 0.0001$. $n = 3$ biological replicates. Dotted lines inside the violin plots show the quartiles and dashed lines depict the medians. Dotted horizontal line marks the zero. **c** Violin-plot showing total quantification of adipocytes per section ($n = 3$, 2–3 sections per animal). Normality was scored as in (**b**). Statistical analysis using Mann–Whitney test showed no significance. Dotted lines show the quartiles and dashed lines depict the median. **d** Barplot showing enrichment test for target GO terms and DEG in WAT. The black box represents differentially regulated genes in WAT of *Mof*$^{+/-}$ animals on SD. Statistical test was performed using the absolute numbers of genes and $p$ values were scored by the two-sided Fisher's exact test. The red bar highlights the only significantly enriched pathway. See also: Supplementary Fig. 6b. **e** Left: Representative immunofluorescence images showing GLUT4 (red) expression in WAT of animals on HFD. Scale bar: 50 μm. Right: MFI quantification of GLUT4 signal. Statistical analysis was performed by two-sided Mann–Whitney test, *$p = 0.048$. $n = 3$ biological replicates. **f** Representative immunofluorescence images showing neutral lipids (BODIPY-493; green) in adipocytes differentiated from mesenchymal stromal cells (MSC). Scale bar: 20 μm. **g** Boxplot showing the quantification of overall lipid droplet area per droplet (μm$^2$) in logarithmic scale (right). Statistical analysis was performed by two-sided ANOVA followed by Tukey post test, *$p = 0.03$, ***$p = 0.0007$. $n = 3$ biological replicates. **h** Graphical scheme illustrating the pre-adipocyte (iAdipo) in vitro differentiation and induction of *Mof* knockout (*Mof*-iKO) by treatment with 4-hydroxy-tamoxifen (4-OHT). See also: Supplementary Fig. 6e. **i** RT-qPCR analyses of iAdipo cells with (+) or without (−) insulin/glucose challenge. Violin plot shows average *Glut4* mRNA expression relative to *Hprt* of biological replicates ($n = 5$). Statistical analysis was performed by two-way ANOVA followed by Holm–Sidak's comparison test, *$p = 0.05$. Dotted lines show the quartiles and dashed lines depict the medians. **j** Representative histogram showing glucose uptake capacity after insulin/glucose challenge (left). Gray histogram represents the untreated cells, control (black), and *Mof*-iKO (purple). Floating plots showing the 2-NBD ratio uptake (treated MFI/ untreated MFI). "+" indicates the treatment employed and the dashed horizontal line marks the basal glucose uptake. Statistical analysis was performed by one-way ANOVA followed by Tukey's comparison test, *$p = 0.013$, ***$p = 0.0001$. $n = 5$ biological replicates. **k** Left panel: Line-plots showing the percentage of extracellular acidification rate (%ECAR) of control (black) or *Mof*-iKO (purple) iAdipo cells without (open circles) or upon insulin challenge (filled circles). Arrows indicate the addition of inhibitors. Statistical analysis was performed by two-sided two-way ANOVA followed by Holm–Sidak's comparison test. Right panel: Violin plot showing quantification overall glycolysis. "+" indicates insulin treatment. Biological replicates ($n = 3$). Statistical analysis was performed by two-sided two-way ANOVA followed by Holm–Sidak's comparison test, **$p = 0.01$, ns = not significant. Dotted lines show the quartiles and dashed lines depict the medians. See also: Supplementary Fig. 6.

the whole transcriptional network regulating glucose import. ChIP-qPCR in WAT detected no enrichment of MOF at the promoters of *Glut4, Pgc1α* or *Mef2c* (Fig. 6h), suggesting that expression of these genes is not directly regulated by MOF. Conversely, the gene coding for PPARγ (*Pparg*), the major regulator of adipocyte biology and glucose homeostasis[79], was downregulated in both *Mof*$^{+/-}$ WAT and glucose/insulin-challenged *Mof*-iKO cells (Fig. 6 and Supplementary Fig. 7i, j). In contrast to *Glut4, Pgc1α*, and *Mef2c*, MOF was bound to the promoter of the main isoform of *Pparg* and this binding was significantly reduced in *Mof*$^{+/-}$ WAT (Fig. 6h). Moreover, using the *Mof*-iKO model, we confirmed that MOF and H4K16ac at the *Pparg* promoter increased upon glucose and insulin treatment (Fig. 6i). We conclude that expression of *Pparg* via MOF-mediated deposition of H4K16ac regulates *Glut4* expression in response to insulin signaling upstream of the transcriptional activators *Pgc1α* and *Mef2c* (Fig. 6g). Intriguingly, we observe a specific effect on the glucose homeostasis transcriptional network rather than a global deregulation of all PPARγ target genes in *Mof*$^{+/-}$ WAT (Supplementary Fig. 7l, m).

Finally, treatment of *Mof*-iKO adipocytes with thiazolidinedione (TZD; a potent PPARγ agonist widely used to treat T2D patients[80]) upon insulin/glucose challenge partially rescued lipid storage defects in *Mof*-depleted cells (Fig. 7a) and restored the expression of *Pgc1α* and *Glut4* (Fig. 7b, c). Of note, TZD treatment had no effect on bulk H4K16ac levels in these cells (Fig. 7d). siRNA-mediated knockdown of *Glut4* in wild-type cells phenocopied lipid storage defects observed in *Mof*-iKO cells, as expected (Fig. 7e–h). In contrast to *Mof*-iKO cells, however, TZD treatment was not able to rescue lipid storage in *Glut4* knockdown cells (Fig. 7e–h).

Together our data show that MOF orchestrates glucose uptake and subsequent lipid storage by transcriptional regulation of *Pparg*. In the absence of a MOF-elicited PPARγ response, the activation of the downstream transcriptional network mediating glucose uptake cannot be triggered (Fig. 7i). The lack of glucose influx, in turn, results in reduced neutral lipid storage in

adipocytes. We conclude that failure in glucose uptake in adipose tissue is a major consequence of *Mof* haploinsufficiency, resulting in resistance to HFD-induced obesity in these animals (Fig. 8).

## Discussion

Our work identifies a fundamental role for MOF in regulating systemic metabolic homeostasis. Using organ-specific metabolomics, we identify global destablization of carbohydrate and amino acid metabolism in the serum and multiple tissues of *Mof* haploinsufficient mice. These metabolite disruptions predispose *Mof*$^{+/-}$ mice to developing metabolic disorders, as confirmed by physiological indicators including impaired glucose assimilation, reduced insulin secretion and elevated glycated-hemoglobin (Hb1ac) levels in these animals. When challenged with prolonged HFD stress, *Mof*$^{+/-}$ mice showed surprising resistance to fat mass gain which prevented them from developing obesity. Adipose tissue of HFD-challenged *Mof*$^{+/-}$ mice is devoid of neutral lipid storage as a consequence of impaired glucose uptake in this tissue. Molecularly, we established that MOF-mediated H4K16ac is a master regulator of *Pparg* expression and is thereby responsible for sustaining glucose uptake and lipid storage in adipocytes. Therefore, we provide the first experimental evidence of a direct causal relationship between a KAT, glucose homeostasis, lipid deposition and obesity.

**Chronic *Mof* depletion promotes a pathological insulin-glucose balance.** The resistance of *Mof*$^{+/-}$ mice to gaining weight upon HFD stress represents a double-edged sword. Despite the apparent benefit in ameliorating obesity, this phenotype occurs at the expense of normal central carbon and amino acid metabolism. These metabolic alterations are, in turn, associated with defective glucose sensing and failed glycaemic homeostasis.

Insulin-dependent glucose homeostasis depends on the interplay between multiple tissues. In a healthy individual, spikes in glucose levels are rapidly sensed by pancreatic β cells, promoting insulin secretion. As a consequence, insulin reaches tissues such

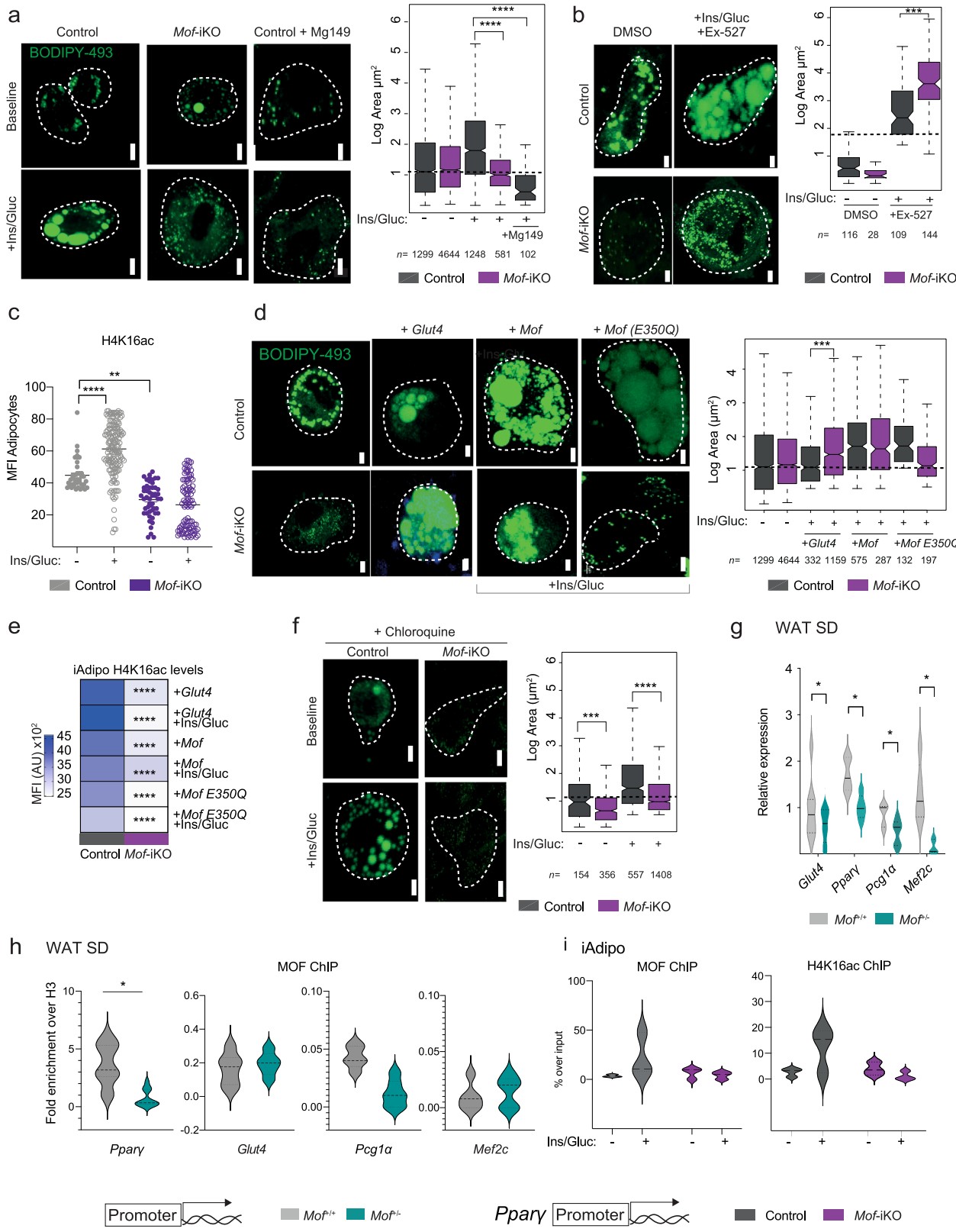

as WAT or SKM, triggering expression and translocation of GLUT4 to boost glucose uptake and reduce extracellular glucose levels. Insulin release is transient and ceases immediately once basal glucose levels are reinstated. We propose a model in which insulin secretion in $Mof^{+/-}$ animals is not completely blocked,

but strongly impaired upon glucose challenge, arguing that glucose sensing is the major defect in these animals (Fig. 8). If glucose is not appropriately sensed in β cells, serum glucose levels will remain high. Stimulating glucose uptake by tissues via insulin signaling represents a potential coping mechanism against this

**Fig. 6 MOF regulates fat storage by regulating the *Glut4* transcription network. a, b, d, f** Left: representative images of neutral lipid staining (BODIPY-493; green). Scale bars: 5 μm. Right: Boxplots showing quantification of overall lipid droplet area per droplet (μm$^2$) in logarithmic scale with the boxes depicting interquartile range. Number of biological replicates per condition: $n = 3$. Total number of droplets used for quantification are indicated in the figure. Dashed horizontal lines mark the mean area of controls. **a** Control, *Mof*-iAdipo or Mg149-treated (50 nM) control cells at baseline ("-", upper panel) or after 15 min of insulin/glucose challenge ("+", lower panel). Statistical analysis was performed using the raw data and one-way ANOVA followed by Kruskal–Wallis comparison test, ****$p = 10^{-16}$. **b** Control and *Mof*-iAdipo cells treated with Ex-527 (200 nM) after insulin/glucose challenge. Statistical analysis was performed using the raw data and one-way ANOVA followed by Kruskal–Wallis comparison test, ***$p = 0.006$. See Supplementary Fig. 7a for baseline images. **c** Scatter-plot depicting the H4K16ac MFI in control and *Mof*-iKO iAdipo with (open circles) or without (filled circles) insulin and glucose challenge. Each dot represents a single iAdipo. Statistical analysis was performed using the raw data and two-sided one-way ANOVA followed by Kruskal–Wallis comparison test, **$p = 0.014$, ****$p = 10^{-16}$. Number of biological replicates $n = 4$. **d** Control and *Mof*-iAdipo cells at baseline, following overexpression of *Glut4*, or following ectopic expression of wild-type *Mof* (wt-*Mof*) or *Mof* catalytic mutant (*Mof*-E350Q) challenged with insulin/glucose (+Ins/Gluc). Statistical analysis was performed using the raw data and two-sided one-way ANOVA followed by Dunn's multiple comparison test, ***$p = 0.001$. Note that the data for untreated wild type ($n = 1299$) and *Mof*-iKO ($n = 4644$) is identical to the data depicted in (**a**). **e** Heatmap showing the H4K16ac levels from wild type and *Mof*-iKO upon ectopic expression of *Glut4*, wt-*Mof* or *Mof*-E350Q with or without glucose and insulin treatment. Statistical analysis was performed using the raw data and two-sided two-way ANOVA followed by Sidak multiple comparison post test, ****$p = 10^{-16}$. **f** Control and *Mof*-iAdipo cells treated with chloroquine at baseline or upon insulin/glucose challenge. Statistical analysis was performed using the raw data and two-sided one-way ANOVA followed by Dunn's multiple comparison test, ***$p = 0.0001$, ****$p = 10^{-16}$. **g** RT-qPCR analyses of visceral WAT. Violin plots show the mRNA expression of *Glut4*, *Pparg*, *Pcg1α* and *Mef2c* as an average of biological replicates (*Mof*$^{+/+}$ $n = 5$; *Mof*$^{+/-}$ $n = 5$ for *Glut4*, *Pparg*, *Pcg1α* and *Mof*$^{+/+}$ $n = 3$; *Mof*$^{+/-}$ $n = 3$ for *Mef2c*). Statistical analysis was performed by two-sided Mann–Whitney test, *$p = 0.026$. Dotted lines show the quartiles and solid lines depict the medians. **h** ChIP-qPCR analyses of MOF at promoters of *Pparg*, *Glut4*, *Pcg1α* and *Mef2c* promoters in visceral WAT. The data are expressed as fold change over H3. Violin plots show the average of biological replicates (*Mof*$^{+/+}$ $n = 6$ and *Mof*$^{+/-}$ $n = 5$). Statistical analysis was performed by two-sided Mann–Whitney test, *$p = 0.015$. Dotted lines show the quartiles and dashed lines depict the medians. **i** ChIP-qPCR analyses of MOF (left) and H4K16ac (right) at promoters of *Pparg* in the presence or absence of Glu/Ins in wild-type and *Mof*-iKO adipocytes. Dotted lines inside the violin plots show the quartiles and dashed lines depict the medians.

---

persistent hyperglycemia. *Mof*$^{+/-}$ animals show a hyper-insulin response in the pancreas, WAT, SKM, and heart. However, the ability of intracellular insulin signaling to produce the desired attenuating effect on serum glucose levels is severely hampered by inadequate levels of the GLUT4 transporter in the membranes of these cells.

Recently, pancreas-specific *Mof* depletion was reported to promote expansion of β cells at the expense of α-cells, ultimately leading to reduced glycemia upon glucose challenge[81]. Surprisingly though, insulin tolerance tests from the same study indicated significant resistance to regulating blood glucose levels upon insulin treatment. This impaired response agrees with our observations that chronic MOF reduction causes impaired glucose sensing due to downregulation of glucose transporters. An independent study has shown that *Mof* deletion in the mediobasal hypothalamus reduces hypothalamic polysialic acid (PSA), resulting in overfeeding and exacerbated diet-induced obesity[82]. In contrast, *Mof*$^{+/-}$ mice show normal feeding behavior and lean mass gain equal to littermate controls. We find that the resistance of *Mof*$^{+/-}$ mice to diet-induced obesity is a direct result of the failure of MOF-mediated glucose uptake in adipose tissue. Furthermore, given the tissue-specific functions of MOF and H4K16ac it is possible that while altered PSA levels are the primary defect upon complete knockout in the hypothalamus, the haploinsufficient model reveals more systemic effects of *Mof* reduction.

**MOF regulates glucose uptake in adipocytes via the insulin-dependent glucose transporter network.** We established that MOF orchestrates glucose uptake and subsequent neutral lipid storage by regulating the transcriptional network upstream of *Glut4* in adipocytes, in particular by binding and promoting expression of *Pparg*. Remarkably, both *Mof*$^{+/-}$ and *Pparg*$^{+/-}$ mice are protected against fat mass gain when fed a HFD[83]. The phenotypic similarities between the *Pparg* and *Mof* haploinsufficient mouse models strongly support a functional interaction between MOF and the downstream PPARγ transcriptional axis in adipose tissue.

Treatment with the PPARγ inhibitor TZD partially restores the expression of *Pgc1α*, *Mef2c*, and *Glut4* as well as lipid storage in *Mof*-depleted adipocytes. Given that TZD rescue is incomplete, it remains unclear whether pathways such as increased thermogenesis or browning of adipose tissue may additionally be contributing to obesity resistance in *Mof*$^{+/-}$ mice. MOF ChIP-seq revealed that *Mof*$^{+/-}$ WAT retained ~5% of mapped gene targets found in *Mof*$^{+/+}$ WAT, indicating that MOF loss likely affects multiple other pathways beyond glucose sensing.

Another intriguing question is whether MOF depletion alters the microbiome. Gut microbiota are known to exert a critical role in metabolic diseases including obesity, partially by eliciting local immune reactions[84]. Considering that the hepatic microenvironment of *Mof*$^{+/-}$ mice promotes Th17 cell polarization in the liver, it would be important to test whether a similar immune response occurs in the gut.

**Altered MOF levels may be implicated in noncommunicable diseases.** Our systematic metabolomic analysis showed differential expression of biomarkers for multiple NCDs following *Mof* depletion, indicating that diabetic predisposition is only one potential angle of interrogating this data. For instance, *Mof*$^{+/-}$ hearts show deregulation of glyoxylate/dicarboxylate, aminoacyl-tRNA biosynthesis and vitamin B6 metabolism, which have all been associated with coronary heart disease[85–87]. Consistent with this finding, mice with conditional *Mof* knockout in the heart succumb to heart failure and premature death[25]. Our analysis also revealed deregulation of vitamin B6 metabolism in the brains of *Mof*$^{+/-}$ mice. Vitamin B6 serves as an important cofactor for many metabolic processes such as protein folding, amino acid biosynthesis, and glycogen degradation[88]. Perturbations in vitamin B6 are associated with several pathologies including autoimmune inflammatory disorders[89] and stroke[90]. These heart- and brain-specific metabolic disturbances suggest that MOF depletion might also facilitate the onset of other, more tissue-localized NCDs.

**MOF in human metabolism.** Taken together, this work identifies the lysine acetyltransferase MOF as a key regulator of glucose and

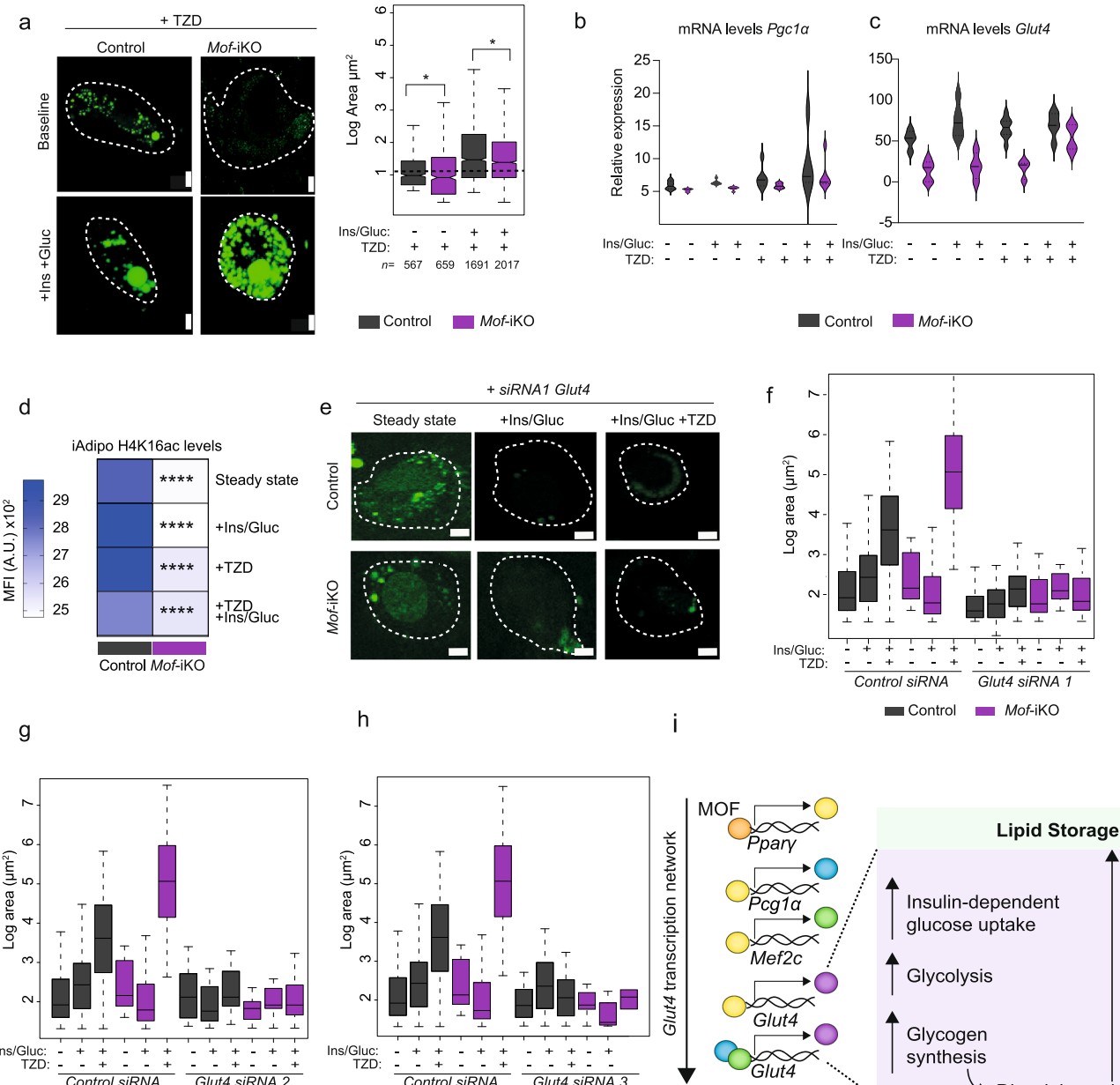

**Fig. 7 PPARγ agonists can rescue lipid storage in *Mof*-iKO adipocytes only in the presence of *Glut4*. a** Left: representative images of neutral lipid staining (BODIPY-493; green). Control or *Mof*-iAdipo treated with thiazolidinedione (TZD)($10^{-4}$ mol/L) at baseline (upper panel) or upon insulin/glucose challenge (bottom panel). Scale bar: 5 μm. Right: boxplot showing quantification of overall lipid droplet area per droplet (μm²) in logarithmic scale with boxes showing the interquartile range. Number of biological replicates per condition: $n = 3$. Total number of droplets used for quantification are indicated on the figure. Dashed horizontal lines mark the mean area of controls. Statistical analysis was performed using the raw data and one-way ANOVA followed by Kruskal–Wallis comparison test, *$p = 0.039$. **b**, **c** RT-qPCR analyses of control or *Mof*-iAdipo cells treated with TZD at baseline or upon insulin/glucose challenge. Violin plots show the average *Pgc1α* (**b**) and *Glut4* (**c**) mRNA expression of biological replicates ($n = 3$). Dotted lines show the quartiles and solid lines depict the medians. **d** Heatmap showing H4K16ac levels after TZD treatment at different conditions. Statistical analysis was performed using the raw data and two-sided two-way ANOVA followed by Sidak multiple comparison post test, ****$p = 10^{-16}$. **e** Representative images of neutral lipid staining (BODIPY-493; green). Control or *Mof*-iAdipo treated with *Glut4* siRNA in the presence or absence of thiazolidinedione (TZD)($10^{-4}$ mol/L) at baseline or upon insulin/glucose challenge (+Ins/Gluc). Scale bar: 5 μm. **f**–**h** Boxplots showing quantification of overall lipid droplet area per droplet (μm²) in logarithmic scale. Dashed horizontal lines mark the mean area of controls. Number of biological replicates per condition: $n = 3$. The "*Control siRNA*" data is identical in (**f**), (**g**) and (**h**). (**i**) Schematic representation of proposed MOF-mediated regulation of the *Glut4* transcription network. *See also*: Supplementary Fig. 7.

amino acid metabolism and adipose tissue homeostasis. Our findings have important clinical implications for understanding the contribution of epigenetic regulators to metabolic disorders such as obesity, cardiovascular disorders, and T2D as well as cancer. Importantly, alterations in MOF levels can serve as an important biomarker to predict future risk of developing metabolic syndromes. We envision that advances in cell-type-specific delivery of small molecule inhibitors aimed at modulating acetylation levels could be an exciting avenue for future research in the battle against obesity and its comorbidities.

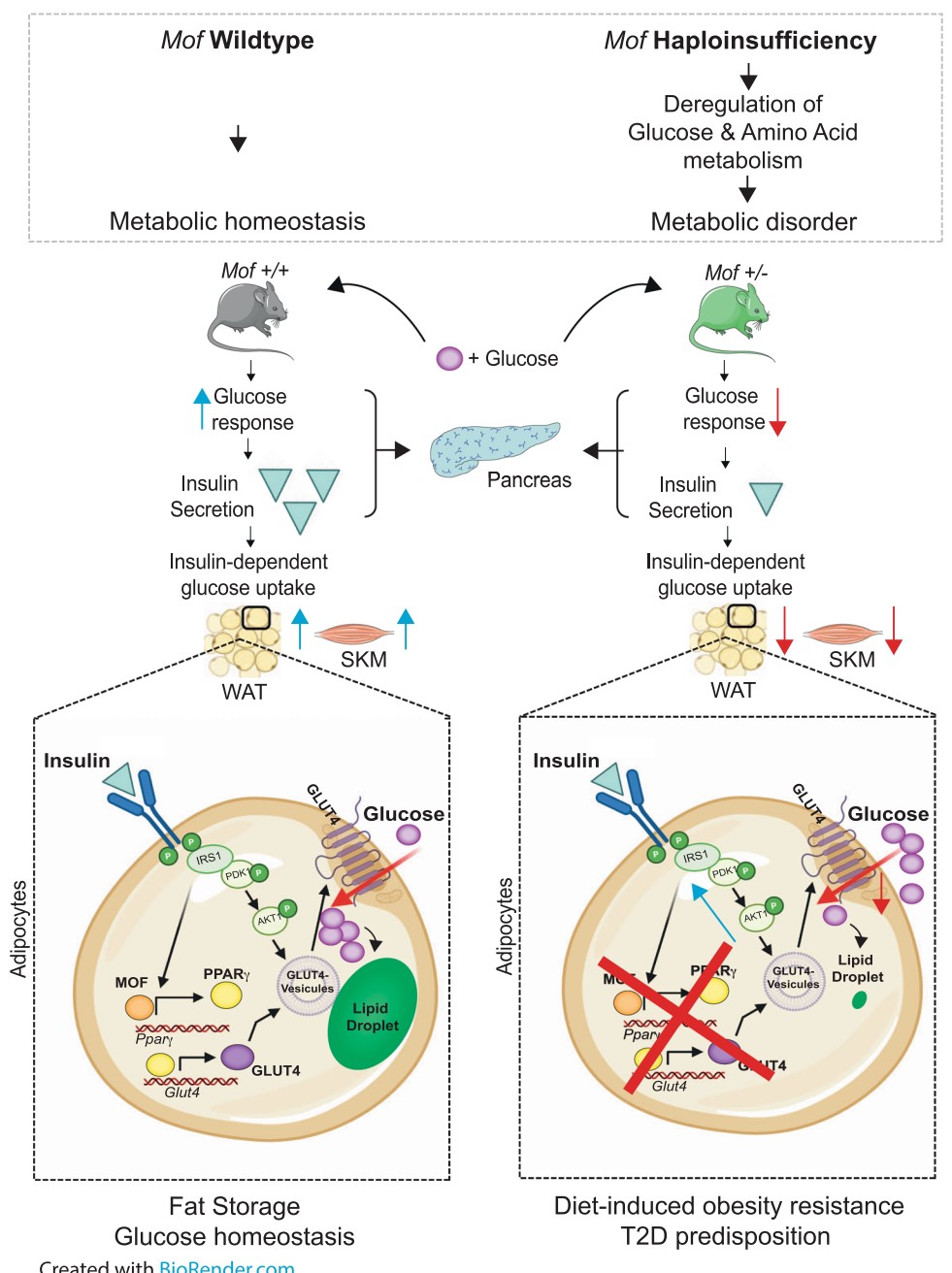

Created with BioRender.com

**Fig. 8 *Mof* Haploinsufficiency Protects against Diet-induced Obesity.** Schematic overview describing the proposed mechanism by which chronic reduction of MOF results in global remodeling of metabolism toward diabetic predisposition. In wild-type cells, insulin binds to the insulin receptor, which will trigger a cascade of phosphorylation events culminating in the phosphorylation of GLUT4 inside storage vesicles. These will be transported to the cytoplasmic membrane to regulate glucose import. In parallel, insulin triggers the transcription of *Glut4* predominantly through the transcriptional network downstream of PPARγ and PGC1α. In healthy adipose tissue and skeletal muscle (SKM), MOF mediates glucose uptake by positively regulating gene expression of the nuclear hormone receptor *Pparg*, thereby triggering the downstream transcriptional network resulting in increased levels of the major insulin-dependent glucose transporter GLUT4. On the other hand, *Mof* haploinsufficient animals show impaired insulin secretion upon glucose challenge, leading to severe destablization of glucose homeostasis. Whilst insulin signaling and GLUT4 vesicle transport in *Mof*-depleted cells are functional, the transcriptional network regulating *Glut4* expression is impaired, resulting in reduced glucose import. Limitation of intracellular glucose is responsible for impaired neutral lipid storage in *Mof*$^{+/-}$ adipocytes. This study identifies the epigenetic regulator MOF as the first histone acetyltransferase to regulate the onset of diet-induced obesity.

## Methods

### Cell culture

*Primary mouse embryonic fibroblast culture.* Primary MEF cells were isolated from embryos of wild-type and Cre-ERT2tg *Mof*$^{flox/flox}$ (*Mof*-iKO) mice at 13.5–15 days after mating. Embryos were removed from the uterus and shortly rinsed with 70% EtOH and ice-cold PBS. The chorion, placenta, heart, liver, yolk sac, and head were dissected out from the embryo. The remaining body was transferred onto a new dish and minced finely. Afterwards, 1 ml Trypsin/1 mM EDTA was added and incubated for 30 min at 37°C. Then, Dulbecco's modified Eagle's medium (DMEM, Life Technologies, 31966047) supplemented with 10% FCS and 1% penicillin–streptomycin (DMEM-10%) was added and the sample was centrifuged for 5 min at 1000 × *g*. The pellet was resuspended in DMEM, plated in 10-cm petri-

dishes and cultured at 37°C with 5% CO2 in a controlled incubator. Confluence was reached after 2–3 days in culture and part of the cells were frozen as passage 1, while the rest were used for MEF expansion and genotype confirmation.

*In vitro differentiated adipocyte culture.* Primary MEFs were differentiated to adipocytes using the protocol of Fei et al.[71] with minor modifications. In brief, $1 \times 10^6$ MEFs (passage 3 or 4) were plated in six-well plates in DMEM-10% supplemented with 0.5 mM methylisobutylxanthine (Sigma, I7018), 1 M dexamethasone (DEX), 10 g/ml insulin and 10 M troglitazone (Tocris Bioscience, 3114/10). This media was kept for 3 days, and then replaced with DMEM-10% for another 2 days. After 5 days in culture 4-OHT was added to the culture twice for *Mof* deletion. For downstream analysis, cells were harvested before differentiation (MEFs) and after differentiation (iAdipo) followed or not by *Mof* deletion.

*PANC-1 cell culture.* Human pancreatic cancer cell line PANC-1 was obtained from the cell line depository of the BIOSS excellence cluster (Freiburg, Germany) and cultured in DMEM containing 10% FCS (DMEM-10%), at 37 °C with 5% CO$_2$ in a controlled incubator.

*In vitro differentiated pancreatic islet culture.* PANC-1 cells were differentiated into ductal and islets-like cells following published protocol[91,92]. In brief, PANC-1 were seeded at $1 \times 10^5$ cells/ml density in 12-well plates with DMEM-10% supplemented with 1% BSA, transferrin (Sigma, 5.5 µg/ml) and insulin-like growth factor 1 (10 ng/ml, PeproTech) for a period of 7 days. Media was changed every other day and differentiation confirmed by qRT-PCR assays.

*Liver organoid formation and maintenance.* Livers from 26-week-old HFD-treated *Mof*$^{+/+}$ and *Mof*$^{+/−}$ animals were harvested and placed in cold PBS. Ductal liver organoids were generated according to published protocol[93] with modifications. In brief, livers were minced thoroughly and washed three times in cold wash buffer (GlutaMAX, Gibco #35050-038, 1% FCS, 100 U/ml penicillin, and 100 µg/ml streptomycin (Gibco #15140-122)) to remove fat. Tissue was digested in pre-warmed digestion media (wash buffer, 0.25 mg/ml Collagenase (Sigma Aldrich #C2674), 0.125 mg/ml (Life Technologies #17105-041)) at 37 °C on a rotating wheel for 1 h. Tissue was pelleted, supernatant discarded and 1 h digestion was repeated using fresh digestion buffer. Ductal structures were pelleted and washed three times in cold wash buffer. Ductal fragments were handpicked and seeded into Matrigel® (Corning, #356231) domes (50 fragments per 50 µl of Matrigel®). Upon solidification of the basement matrix, domes were overlaid with isolation medium and incubated for 3 days under standard cell culturing conditions (37 °C with 5% CO$_2$) after which media was exchanged to expansion media[93]. Ductal organoids were maintained and a 14 day hepatocyte differentiation protocol was performed as described in ref. [93].

## Liver organoid and CD4+ T cell co-culture

*Fluorescence-activated cell sorting.* For co-culture, wild-type thymi were isolated and thymocytes retrieved by mechanical filtering on a 70-µm cell strainer. The cells were resuspended in PBS, centrifuged at $500 \times g$ for 5 min and the pellet was resuspended in ACK lysing buffer for erythrocyte cleaning. After 10 min, PBS was added to the cellular suspension followed by centrifugation. The remaining leukocytes were resuspended in FACS buffer containing zombie-dye for dead cells, fluorophore-conjugated primary antibodies against CD3 (PERCP-Cy5.5), CD4 (A700), and CD8a (PE) and kept on ice for 30 min. Afterwards, cells were resuspended in a 1x v/v FACS buffer and centrifuged at 3000 g for 30 s. The pellet was resuspended in FACS buffer and naïve CD8a+ cells sorted using the BD Aria-FACS Fusion II using a nozzle of 70 µm and maximum flow of 15,000 events/s rate. A 95% pure population was sorted and used for co-culture.

*Organoids and naïve Th cell co-culture.* Sorted purified naïve T cells were then co-culture with differentiated hepatic organoids at a ratio of 2:1 (liver organoids: T cells) in the presence of IL-2 (10 ng/mL, Peprotech 212-12). T cells were further separated into two groups (a) stimulated with CD3:CD28 dynabeads (Thermo Fisher, 11456D) in a 1:1 ratio or (b) no stimuli. Five days prior to the co-culture experiment differentiated hepatic organoids were cultured in AIM V™ media (Thermo Fisher) supplemented with 3 µM DEX. After 5 days of co-culture, cells were harvested and prepared for flow cytometry analysis.

For cytokine flow cytometry evaluation we added 1X Brefeldin-A for at least 10 h prior to analysis. The cells were then transferred to 1.5 ml tubes and centrifuged at $3000 \times g$ for 30 s. The remaining pellet was washed 2× with FACS buffer and fluorophore-conjugated primary antibodies against CD4 (PE), and viability dye (zombie-dye aqua or green) was added and incubated for 30 min incubation on ice. Then the sample was centrifuged and pellet washed twice with 1× wash buffer (Foxp3/Transcription Factor Staining Buffer Set, Affymetrix, eBioscience, USA, 00-5523-00), followed by 40 min incubation on ice with the "Perm/Fix" solution (1:4 dilution) (Foxp3/Transcription Factor Staining Buffer Set, Affy- metrix, eBioscience, USA, 00-5523-00). Samples were washed once with 1× wash buffer and twice with FACS buffer followed by incubation with fluorophore-conjugated primary antibodies against IFN-gamma (FITC), IL-17A (A647), RORgammat (PERCP-Cy5.5), FoxP3 (AF700), and/or IL-4 (BV421), IL-10(BV421)

as indicated in the figures. The cells were incubated for ~16 h, washed twice with FACS buffer and immediately acquired using the BD Fortessa II or Fortessa I cytometer. Data were next analyzed using the FlowJo v.10 software.

**Animal husbandry.** Animals were maintained on a 14/10-h light–dark cycle under controlled humidity and temperature. For SD experiments the animals were provided with a normal chow diet 15% fat (Ssniff GmbH), while obese animals were fed via HFD (60% kcal/% fat, Research Diet). Both groups were fed ad libitum, had free access to water and refinement methods as nesting material were employed. The gain of total body mass was recorded every second week. Every mouse strain in this study was backcrossed with C57BL/6J mice. All animal procedures are in agreement with and approved by the responsible Animal Welfare Committees (Regierungspräsidium Freiburg, Karlsruhe/ Germany, license, G-17/150), conforming to the Guide for the Care and Use of Laboratory Animals published by the National Institutes of Health (publication No. 85-23, revised 1996).

## Mouse phenotypic assays

*Oral glucose tolerance test and plasma insulin measurements.* For the OGTT, mice were fasted overnight (16 h), before starvation the total body weight, fat, and lean mass was recorded as well as free and total water contribution using the EchoMRI™ scan. Then, after 16 h the animals were again scanned using the EchoMRI™ scan and basal blood glucose was measured by Accu-check stripes and the OneTouch Vita blood glucose meter. Mice were given glucose bolus (1 g/kg body weight) by oral gavage. Blood glucose levels were measured using a OneTouch Vita blood glucose meter at −5 (to avoid stress bias), 0, 15, 30, 45, and 60 min after glucose. Blood drawn using heparinized capillary tubes was centrifuged at $2000 \times g$ for 15 min at 4 °C. Plasma obtained was used for insulin level measurement by ELISA (Mercodia Ultrasensitive Mouse Insulin Kit).

*Glycated-hemoglobin measurement.* For glycated-hemoglobin (HbA1c) animals were euthanized by asphyxiation and blood was taken using EDTA-coated syringes. Whole blood was used and HbA1c levels measured by colorimetric assay using hemoglobin A1c human kit (Crystal Cheme, 80099).

## Histological preparation

*Morphologic analysis—Hematoxylin & Eosin staining.* Queried tissue was dissected, washed in cold PBS, and incubated for 24 h at 4 °C in 4% paraformaldehyde methanol-free solution. Samples were dehydrated stepwise by incubation in EtOH solutions of 70, 80, 90, and 100%. In each solution samples were incubated at 4 °C for 24 h. Dehydration was completed by 30 min incubation in Histolemon-Erba (Carlo-Erba, 8028-48-6), followed by paraffin embedding [formaldehyde-fixed paraffin embedding]. Paraffin blocks were sectioned with a RM2155 microtome (Leica) into 5- to 7-µm slices. Liver sections were washed with ice-cold PBS and fixed in ice-cold methanol for 20 min on ice. The specimens were then stained with hematoxylin solution [0.1% hematoxylin, 5% KAl(SO4)2, and 0.02% KIO3]. Counterstaining was performed by incubating the slides in Eosin solution (1% eosin). Tissue imaging was conducted using the Apotome-I (Zeiss) brightfield microscope.

*Immunofluorescence stainings and image acquisition.* WAT, liver, and spleen, whole organs were dissected, weighed, and embedded into OCT (Tissue-Tek, SA62550-01) and frozen gradually: first the blocks were left on a dry-ice board for 15 min followed by liquid nitrogen submersion. OCT blocks were sectioned with a cryotome (Leica) at −20 °C into 7–10-µm slides.

For BODIPY staining, WAT specimens were blocked in 2% calf serum PBS-T for 15 min at RT before incubating with the BODIPY probe, BP (Invitrogen). Each slide was rinsed two times with PBS-T, for 5 min each. Specimens were incubated with BP diluted in PBS (1:300) for 30 min at 37 °C in the dark. After rinsing with PBS-T, VectorShield with DAPI and coverslip were mounted. While for GLUT4 staining WAT sections were embedded in paraffin as describe for H&E staining, sections were deparaffinized and rehydrated stepwise using an EtOH gradient (2 × 100% xylene 5 min each, 2 × 100% ethanol 5 min each, 2 × 95% ethanol 5 min each, 70% ethanol for 5 min) and rinsed in dH$_2$O for 5 min.

For liver and spleen images, cryosections were fixed in cold methanol for 20 min on ice followed by 3x PBS-T wash, 10 min each. Samples were then blocked with a 2% BSA-PBS-T solution for 2 h at RT. After, specimens were washed as above and liver sections were incubated with the Phalloidin-A647 probe for 30 min at RT. Then liver samples were washed 3× in PBS-T and mounted as above. While spleen sections were incubated with fluorophore-conjugated primary antibodies against IgD (FITC, eBioscience, 115993), F4/80 (BV421,BD 565411), CD11c (APC, BD 550261), and propidium iodide for 12 h at −4 °C. Specimens were subsequently washed 3× with PBS-T and samples were mounted with pure VectorShield. Images were acquired using either the LSM780 or Airyscan (Zeiss) and analyzed using the ImageJ software. Images were taken using the ×40 magnification.

*Oil red O and BODIPY staining in adipocytes.* Oil red O staining was performed 5 days after the adipocytes induction. In which cells were washed twice with cold PBS and fixed with 1% methanol-free paraformaldehyde in PBS for 30 min at RT. Then the well was washed 2× with PBS, 3× with water, one time with 60%

isopropyl alcohol, and stained with oil red O (six parts of 0.6% oil red O dye in isopropyl alcohol and four parts water) for 2 h. The excess stain was removed by washing with ice-cold 60% isopropanol and 2× with water. The plate was then observed under an inverted-brightfield microscope. BODIPY staining followed the same protocol as the one performed for WAT.

*Image quantifications.* For all morphometric analysis of adipocytes, islets, hepatocytes, and spleen follicles 3–5 animals of each genotype and food group were analyzed. Briefly, individual channels were converted to 8-bit grayscale and measurement scale was converted from pixels to mm. An identical threshold was applied to all images from the same channel to exclude background signals and watershed was applied as a mask. The neutral lipid storage was measured using the ImageJ macro "Lipid droplets Tool". Images were acquired using the 40× objectives and a threshold of 1 μM settled for droplets identification.

## Metabolomics

*Mice used.* Brain, heart, liver, kidney, spleen, and serum samples-
Mice used: 8-week-old $Mof^{+/-}$: four mice used in total, two males and two females. 8-week-old $Mof^{+/+}$ control mice used: three mice used in total, one male and two females

WAT sample- Mice used: 8-week-old $Mof^{+/+}$ and $Mof^{+/-}$: 3 males of each genotype.

*Sample collection.* Mice were sacrificed by cervical dislocation and all tissues were harvested in the same chronological order for every animal. Excess blood was washed off by a quick rinse in freshly prepared ammonium carbonate wash solution (75 mM, pH 7.4). Each tissue was rapidly dissected into 2–4 slices, on ice. WAT samples were collected in 0.5 mL PBS on ice. Samples were flash frozen and stored in −80 °C until metabolite extraction.

*Polar metabolite extraction from brain, heart, liver, kidney, and spleen.* Approximately 50-mg tissue slice was used per extraction. Tissue homogenization was performed in Milli-Q water pre-heated to 80 °C (1 ml/ 50 mg tissue) with metal bead-based homogenizer—Retsch mixer mill MM 400. Lysates were incubated at 80 °C for 3 min in a thermomixer, before being spun down at 7000 g for 3 min at 22 °C. The pellets were used for protein estimation. Supernatant containing metabolites was transferred to a fresh tube and stored −80 °C, till FIA-MS analysis (U.Sauer Lab, ETH Zurich).

*Polar and apolar metabolite extraction from WAT.* Thirty milligrams tissue was homogenized by mechanical disruption as mentioned above. Polar and apolar metabolites were extracted by adding 400 μl of cold methanol and 400 μl cold MiliQ. Samples were then vortex for 30 s. Next, 800 μl of ice-cold chloroform was added and vortex for 3 min, followed by centrifugation at 20,000 × g for 2 min at 4 °C. The polar (top) and the organic phase were separated and transferred to fresh ice-cold 1.5 mL eppendorfs. Universal internal standard mix (ISTD) controls were added for absolute quantification before samples were vacuum dried and stored at −80 °C, till LC–MS/MS analysis (Metabolomics Core, MPIIE Freiburg).

*Serum metabolite extraction.* Minimum 50 μl of blood was retrieved from the mouse heart and placed in a 1.5 ml tube. Then 4× volume of cold ultrapure methanol (MS-grade) was added to the whole-blood specimens and vortexed thrice 30 secs each before incubation at −20 °C for 1 h. Afterward, the samples were centrifuged at 7000 × g for 3 min at RT. The supernatants were transferred to a new 1.5 ml tube and stored at −80 °C till FIA-MS analysis.

## Mouse metabolomics data analysis

*Spectral data processing and ion annotation.* Brain, liver, spleen, kidney, skeleton muscle, and serum mass spectrometry data were processed and analyzed with Matlab (The Mathworks, Natick). Negatively charged ions were tentatively annotated to the Human Metabolome Database (HMDB)[94] based on accurate mass using 0.001 Da tolerance. Importantly, accurate mass does not allow distinguishing between compounds with identical molecular formulas and, hence, ions can match multiple compounds. The WAT dataset was analyzed using the TASQ software using the MPIIE automated pipeline followed by iPath3 of the polar $Mof^{+/+}$ versus $Mof^{+/-}$ comparison.

*Fold change and q value calculation.* Mouse tissue metabolomics data were analyzed through two orthogonal pipelines, which were cross checked to verify robust identification of similar metabolite FC:

(a) MetaboAnalyst software 4.0[95] based analysis was performed based on published protocol from Chong et al.[96]. All tissue datasets were normalized together using quantile normalization. For each tissue set, metabolite FC and associated *p* values were calculated by unpaired *t*-test, by comparing the $Mof^{+/-}$ and $Mof^{+/+}$ sample means. Post hoc calculation of *q* value was performed using the Storey method[97]. Unless otherwise stated, all further analyses were performed on significantly changing metabolites (*q* value < 0.1) identified through this pipeline.

(b) Omu R package[98] based analyses were performed using published protocols.

*Pathway analysis.* All associated HMDB ids for each ionMz were considered for pathway analysis. MBRole2.0[99] software was used to identify enriched modules, pathways and taxonomic terms. Kyoto Encyclopedia of Genes and Genomes (KEGG) (mouse specific) and SMPDP (HMDB id based) databases were used as primary background datasets. Significantly enriched terms had FDR value < 0.05, with minimum four metabolites in set.

*MEBA feature identification.* Multivariate Empirical Bayes Approach (MEBA) for comparative analysis of time series data was performed according to the protocol published by Chong. et al.[96] in MetaboAnalyst software 4.0.

*Metabolite class annotation and tissue enrichment.* Metabolic class was generated by intersection of inchikey from uniquely identified metabolite names with the HMDB metabolite attributes list. The tissue enrichment was scored by constructing a contingency table querying the total number of metabolites, total number of deregulated metabolites and their related class. Metabolic class enrichment was calculated by Fisher.

*Cross-tissue correlation.* To identify significant cross-tissue correlations we subject the log 2 fold change of differential metabolites from each tissue set and construct a matrix containing their correlation coefficient and significance by Pearson correlation. Metabolites with a correlation *p* value < 0.05 in 95% were considered as significant. Calculations were done using R.

*Other data analyses.* Metabolite class annotation was performed using ClassyFire[100] followed by manual curation. Deregulated metabolites Z-Score were plotted as heatmap using the R *pheatmap* package.

*Data analysis and visualization.* Identified and quantified peak intensities were analyzed using MSstats[101]. Peptide intensities from evidence.txt were log-transformed and normalized by the "equalizeMedians" option. Statistical comparison was based on a built-in linear mixed-effect model[102]. which provides fold change estimates and *p* values adjusted to control the FDR level[103] at the 0.05 cutoff. Data were afterward visualized with the use of ggplot2 package[104].

## Transcriptome analysis

*Total RNA extraction.* For RNA extraction from various sources, samples were lysed directly in TRIZOL and homogenized using an electric tissue homogenizer. Following incubation with equal volume of 95% ethanol, they were transferred to a Zymo-Spin™ column (Zymo, R1054). Pure total RNA was extracted according to the manufacturer's instructions. RNA concentrations were quantified on a Qubit 2.0 Fluorometer (Life Technology) and quality for deep sequencing checked by Fragment analyzer.

*Quantitative real-time PCR.* qPCR was performed on a Roche LightCycler® II using FastStart Universal SYBR Green Master (Roche, 04913914001) in a 7 μl reaction at 300 nM final concentration of each primer. Cycling conditions as recommended by the manufacturer were used. Normalization was conducted by *Drosophila melanogaster* total RNA (1:10) spike-ins. Experiments were conducted using at least three independently biological replicates. Primers used in this study are provided in Supplementary Data 5.

*Bulk WAT mRNA-seq.* Equal amount (weight) of visceral white adipocyte tissue (WAT) from 8–10-week-old male mice was used for RNA extraction, using the Total RNA extraction method. Samples were further cleaned and concentrated using the Oligo Clean & Concentrator kit (Zymo, D4061). After quality control, Illumina TruSeq RNA Sample Prep v2 (RS-122-2001) was used for library preparation, and samples were sequenced on Illumina NovaSeq6000 sequencer. All sequencing data were performed in triplicates for the SD group and duplicates for the HFD group. The reads were conducted at 2 × 100 bp length.

*Bulk RNA-seq bioinformatic analysis.* Reads for mouse bulk RNA-seq datasets were mapped following the default SnakePipes parameters of bulk RNA-seq pipeline[105]. In brief, reads were mapped using STAR software alignment against mouse genome version GRCm38. The total number of sequenced reads averaged 15 million pairs of which mean alignment of 65% of the reads were uniquely mapped. Reads were counted with featureCounts (subread-1.5.0-p1). Differential expression analysis was performed with DESEQ2 (v1.26). Multi-FASTA QC statistics indicated data were of high quality and sequencing depth was sufficient to test for differential expression between conditions. Differential genes were called with an FDR threshold of 0.05. For the $Trim28^{+/D9}$ dataset the differentially expressed genes matrix was used as given[15]. Correlations between datasets and graphical output were conducted in an R environment.

*Single-cell RNA-seq and gene regulatory network construction.* Mouse single-cell count matrix from SKM, adipocyte, kidney, pancreas, liver, and heart single-cell

analysis was obtained from the PanglaoDB database[41]. These data were then further processed using the Seurat v3 algorithm, in which standard preprocessing was followed. In brief, we filtered cells that have unique feature counts over 2500 or less than 200 and cells that showed >5% mitochondrial counts. After QC filtering, the data were normalized by employing a global-scaling normalization method that normalizes the feature expression measurements for each cell by the total expression, multiples the results by 10000 and log-transforms the product. Highly variable transcripts were identified using the Seurat3::FindVaribleFeatures[106] function. The data were scaled to provide an equal weight in downstream analysis and buffer the noise of highly-expressed genes. Then, linear dimensional reduction was applied on the scaled data. PCA and "Elbow plot" were used to define the dimensionality of each dataset. Finally, to explore feature expression similarities and defining cell populations we generated the UMAP for each tissue dataset and extracted their differentially expressed genes. After cluster identification and validation that most of the expected organ cell populations were present in the dataset, we used the tissue gene expression matrix to build a GRN using the GENIE3 algorithm. To predict the candidate genes we restrict the candidate regulators to *Mof* and use the "Random Forest" machine learning algorithm. The resulting weight matrix was used to explore gene set enrichment pathways related to *Mof*. Those analyses were performed in a R environment.

*Kyoto Encyclopedia of Genes and Genomes enrichment analysis.* For KEGG enrichment analysis, the list of candidate genes was subjected to the *clusterProfiler()* software. Wherein the function *clusterProfiler::enrichKEGG*[107] with a 0.01 *p* value cutoff was used. Enrichment analyses were conducted in the R environment.

*Other data analyses.* Heatmaps and scatter plots were generated in R with *pheatmap* software and *qplots* default settings. For heatmap representation the data were transformed in *Z*-score prior representation.

## Chromatin immunoprecipitation (ChIP)

*Chromatin preparation and immunoprecipitation.* ChIP was performed according to published protocol[108] with the following modifications: WAT of 8-week-old SD-fed animals was minced and transferred into a dounce homogenizer (loose pestle), covered with 1% formaldehyde in PBS, and dissociated, followed by 15 min incubation at room temperature (RT). During incubation, the tissue suspension was filtered using a nylon 70 µM cell strainer (Falcon, 352350). Excess formaldehyde was quenched by incubation with 125 mM glycine at RT for 5 min and the suspension was pelleted at 500 *g* for 5 min. Cell pellets were washed twice in ice-cold PBS and pelleted for 5 min at 500 *g*. For NEXSON-based nuclei isolation cell pellets were resuspended in 1 ml of lysis buffer (10 mM Tris-HCl pH 8, 10 mM NaCl, 0.2% Igepal, protease inhibitor cocktail). Cell suspension was transferred into Covaris MilliTube (cat. No. 520130) and sonicated in Covaris instrument (E220) for 30 s at peak power 75 W, duty factor 10% and 200 cycles/burst. Nuclei were pelleted at 1000 *g* at 20 °C for 5 min. Pellets were resuspended in 0.5% SDS and incubated at RT for 10 min. Sample was diluted 1:5 in the digestion buffer (1.1% Triton, 1x CutSmart buffer, H2Odest). Up to $1 \times 10^6$ nuclei were digested with Cvik1 (5U/100,000 cells) at 25 °C for 17 h, shaking at 800 *g*. Nuclei were pelleted for 5 min at 1000 *g* and washed in a nuclei wash solution (10 mM Tris-HCl pH 8, 0.25% Triton, 0.2 mg/ml BSA) until residual fat was removed. Nuclei were pelleted at 5000 *g* for 10 min, resuspended in the shearing buffer (10 mM Tris-HCl pH 8, 0.1% SDS, 1 mM EDTA), transferred into Covaris MicroTube (cat. No. 520052) and sonicated for 6 min at peak power 105 W, duty factor 2% and 200 cycles/burst. The sample was diluted 1:6 in 1× iC1b buffer (iDeal ChIP-seq, Diagenode) and incubated with 2 µg of antibody o/n at 4 °C. The next day protein G Dynabeads were added and incubated for 3 h at 4 °C. Beads were washed and DNA was eluted according to manufacturer's instructions (iDeal ChIP-seq, Diagenode). DNA was purified using MinElute PCR purification kit (Qiagen) for further analysis.

*Library preparation.* Libraries were prepared using the NEBNext Ultra II DNA Library Prep Kit for Illumina according to the manufacturer's instructions (New England Biolabs, E7645). ChIP libraries were sequenced with 2 × 50 bp paired-end reads on an Illumina NovaSeq6000 sequencer.

*Processing of ChIP-seq datasets.* ChIP-seq datasets were mapped to GRCm38/Mm10 with default paired-end parameters from snakePipes version 2.1.2[105]. Peaks were called with MACS2 (Galaxy version 2.1.1.20160309.3), and bandwidth was set to 200, lower mfold bound to 5, upper mfold bound to 500, and the *q*-value to 0.1. Input was used as control to call peaks. Peaks from each cell type were merged using cat, BEDtools sort (Galaxy version 2.27.0.0) and piped to BEDtools merge with a distance of 1000 bp. *BamCompare* (Galaxy version 2.5.1.0.0) with a bin size of 50 bp. The data were normalized as Log₂ fold change over H3 immunoprecipitation. Heatmaps were generated using deepTools2 *compute matrix* and *plotHeatmap* function from Deep-Tools2 version 3.5.0[109]. Boxplots were generated using DeepTools2 multi-BigwigSummary scores. The Bioconductor package ChIP-Seeker[110] was used to retrieve the nearest genes around the peak, annotate the genomic region of the peak, and retrieve peak features annotation. GO term analysis was performed using the Metascape[111] platform and the ShinyGo v0.61 database[112].

## Functional tests of in vitro differentiated adipocytes

*Glucose uptake assay.* For measurement of glucose uptake, at least 2000 wild-type and *Mof-iKO* iAdipo cells were cultured in DMEM-10% supplemented with 10% glucose with or without 10 g/ml of insulin with 200 mM of (-n-(7-nitrobenz-2-oxa-1,3-diazol-4-ylamino)-2-deoxyglucose) (2-NBD-Glucose, 2 NBDG) for 30 min. Then cells were washed 2× with PBS and resuspended in FACS-Buffer (0.1%BSA, 1 mM EDTA, PBS) containing DAPI and immediately analyzed by flow cytometry using the BD cytometer Fortessa I/II. The data were then analyzed using the FlowJo v.10 software.

*Differentiated adipocyte (iAdipo) glycolytic capacity analysis.* After *Mof* deletion iAdipo cells were plated in 96-well XF-plates (Agilent) and resuspended in glucose-phenol free media with 2 mM glutamine and incubated for 40 min at 37 °C under deprived $CO_2$ conditions. The plates were assayed on the XFe 96 analyzer, to calculate glycolytic rate by measuring the ECAR over time, after sequential addition of respiratory and glycolytic inhibitors. First, basal ECAR was measured, in the absence of glucose or pyruvate. Then, glucose (10 mM) was added to enhance glycolysis, before inhibiting mitochondrial ATP synthesis (with 1 µM Oligomycin) to estimate maximum glycolytic capacity. Finally, the glucose analog 2-deoxy-glucose (2-DG, 50 mM) was added to inhibit glycolysis. Assay data were exported as a graph pad file and plotting/statistical analysis was performed using the Prism software.

## Knockdown, ectopic expression and chemical treatments

*Mof knockdown in vitro differentiated pancreatic islets and insulin response.* For siRNA knockdowns, 5 nM of Silencer Select (Ambion) control siRNA 4390846 or siRNA s38569 for *MOF* were performed on in vitro differentiated pancreatic islets for 3 days in triplicates using RNAiMAX (Thermo Fisher Scientific). Before knockdown, cells were seeded on 12-well plates in which in the bottom of the well we insert a coverslip. After, silencing cells were then crosslinked with 4% methanol-free formaldehyde in PBS at RT for 10 min and permeabilized with 0.1% Triton-X and 1% BSA in PBS for 30 min at RT. Primary antibody against insulin were diluted (1:100) in FACS buffer and incubated at 4 °C for ~16 h. Wells were then washed 2× with PBS-T and secondary fluorescently labeled antibodies were used to reveal target proteins. DAPI was used to stain DNA. Imaging was performed with the LSM780 confocal microscope.

*Ectopic* Mof *and* Glut4 *expression in in vitro differentiated adipocytes.* The *Glut4* coding sequence was sub-cloned from Addgene plasmid number 52872 and *Mof* coding sequence was cloned into a pcDNATM5/pCMV vector. In vitro differentiated adipocytes were transfected with this vector using LTX Lipofectamine (ThermoFisher, A12621) in a 5:1 Lipofectamine LTX (µl) – to – DNA (µg) ratio. Twelve hours after transfection, the media was changed to the respective cell culture media. Functional experiments were conducted 24 h after the first transfection. Transfection efficiency was determined by pmaxGFP Vector (Lonza, V4XP-3012). pB-GLUT4-7myc-GFP was a gift from Jonathan Bogan (Addgene plasmid 52872; http://n2t.net/addgene:52872; RRID:Addgene_52872)[113].

*Chemical treatments.* After confirming differentiation, iAdipo were exposed to various chemical compounds as indicated in the main text. The following final concentrations were used: Insulin (Sigma, 11061-68-0,100 nM), glucose (Sigma, G8270, 250 nM), chloroquine phosphate (Sigma, 1118000, 100 µM), Ex-527 (TOCRIS, 2780, 200 nM, treatment was performed 24 h prior imaging) and MG149 (12, 25, 50, 100 nM, treatment was performed 24 h prior to imaging) and TZD (10⁻⁴ mol/L). Cells were imaged 15 min after insulin treatment.

## Statistical analysis
All bar plots depicted in this manuscript show the mean value ± standard error of the mean (±SEM) and the solid line depicts the median. All violin plots use dotted lines to show the quartiles and dashed or solid lines to indicate the medians. Boxplots were generated using the boxplot() function in R with the upper whisker = min(max(x)), Q_3 + 1.5 * interquartile range (IQR) and the lower whisker = max(min(x), Q_1 – 1.5 * IQR) where IQR = Q_3 – Q_1, the box length. Notches in boxplots display a confidence interval around the median which is normally based on the median ± 1.58*IQR/sqrt(n). Data were tested statistically using two-tailed or one-tailed tests as indicated in the figure. Normality was scored by the Shapiro-Wilk and D'Agostino Pearson omnibus test. *P* values above 0.05 weeres considered to be statistically significant.

**Reporting summary**. Further information on research design is available in the Nature Research Reporting Summary linked to this article.

## Data availability
All sequencing data from this study have been uploaded to the NCBI GEO database. Raw data pertaining to RNA-seq and ChIP-seq experiments are deposited under the accession number GSE156463. Source data are provided with this paper.

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

## Acknowledgements

We are grateful to M. Diether from U. Sauer Lab (ETH Zurich) for providing us with annotated metabolomics spectral data for brain, heart, liver, kidney, spleen, and serum samples. We are highly obliged to A. Drougard for helping us set up mouse OGTT assays. K. Ganter for preparing some of the histology slides. We thank all the Akhtar lab members for their critical feedback, especially T.H. Tsang (Jose). Our special thanks to J. Büscher, E. Dror, and A.J. Bannister for their critical feedback. The MPIIE core facilities for Metabolomics (especially J. Büscher), Proteomics, Imaging, Deep sequencing, Bioinformatics (especially for W. Deboutte), Mouse and Flow cytometry (especially S. Hobitz and K. Schuldes) have been invaluable for this project. We are also thankful to

J. Seyfferth and H. Holz for their technical help. We are also grateful to K. Diamanti for kindly providing his metabolite analysis. The BioRender Team for their schemes and drawings used in Figs. 4a, 5h, 8 and Supplementary Figs. 2a, 2c and 4n. This study was supported by the German Research Foundation (DFG) under Germany's Excellence Strategy (CIBSS – EXC-2189 – Project ID 390939984). This work was also supported by the German Research Foundation (DFG) under the CRC 992 (A02), CRC 1425 (P04), and CRC 1381 (B3) awarded to A.A.

## Author contributions

C.P.R., A.C. and A.A. conceptualized this study. A.C. and C.P.R. performed the metabolites extraction; A.C. performed the metabolomic analysis; A.C. and C.P.R. conducted downstream metabolomic analysis and data visualization; A.C. and W.S. conducted proteomic sample preparation and analysis; C.P.R. and T.S. performed metabolic profiling of animals and tissue extraction; C.P.R. performed immunostaining, morphometric analysis, cell culture experiments, flow cytometry, biochemical experiments, and conducted the NGS analysis. M.W. and C.P.R. generated the liver organoids; C.P.R. prepared the adipocyte samples for chromatin immunoprecipitation (ChIP). M.W. performed ChIP experiments. C.P.R., A.C. and M.W. wrote the manuscript draft. A.C., C.P.R., and M.W. interpreted the results and prepared the figures. A.C., M.W., C.P.R., M.S., and A.A. wrote the final manuscript version with feedback from all co-authors. A.A. provided supervision and obtained funding for this study.

## Funding

## Competing interests

The authors declare no competing interests.
