## [Peer Review File · Nature Communications]

Histone H4 lysine 16 acetylation controls central carbon metabolism and diet-induced obesity in miceREVIEWER COMMENTS

Reviewer #1 (Remarks to the Author):

In this manuscript, Akhtar and colleagues provide a comprehensive metabolic characterization in MOF (the H4K16 Acetyl transferase) +/- mice, indicating that organismal metabolism is clearly affected by altered levels of MOF. They provide further evidence to indicate cell autonomous effects in the pancreas that affect insulin secretion, and in parallel defined MOF as a direct modulator of PPAR-g in fat cells, in turn influencing glucose uptake and lipid droplets formation in these cells. In vivo, such roles protected MOF +/- mice from high fat diet. Overall, the study includes an enormous amount of data from in vivo and in vitro metabolomics, proteomics, RNA sequencing, ChIP-seq and metabolism studies that has not been performed in the past, and as such it represents useful information to the field. Yet, the manuscript reads as a collection of different findings, with the authors moving from one to the next without much rationale, and in some cases with findings that are not fully explained by the authors' model.

Specific comments

- One major concern relates to the fact that the authors start the paper with a full metabolomics characterization of the different tissues in the MOF hets, that the authors claim they appear much earlier than changes in the proteome. This data suggest that those major metabolic changes are NOT related to transcriptional/translational effects, yet they spend the rest of the manuscript on characterizing direct transcriptional targets of MOF, like PPAR-g. As such, it is unclear why these general metabolic changes occur. The authors may want to move that data to the Suppl. Figures, or else try to explain the molecular reasons for such changes (one possibility: the authors observed major increase in macrophages even in normal diet, suggesting a basal increase in organismal inflammation, which can easily explain the metabolic changes). I do appreciate following such line of studies is well beyond the scope of the study, and hence my suggestion of moving this data to the Suppl. Figures.

- In Figure 1D, the authors claim that changes were observed in specific tissues but not in others, however they analyzed much more of the positive tissues (spleen (n = 380), brain (n = 281), heart (n = 209) and serum 117 (n = 293) showed significant alterations, while liver (n = 22), SKM (n = 36) and kidney (n = 4) exhibit no changes. Can the authors explain why the significant difference in tissues analyzed? Such a difference could easily affect the results.

Extended figure 2A: the authors used Hela cells KD for MOF (adapted on galactose) as supporting evidence for their in vivo metabolomics. These are transformed cells that do not reflect, at all, any relevant tissue, and it is unclear why they included these cells in the manuscript. It will be much more relevant to perform metabolomics in the in vitro differentiated fat cells that they use extensively in the second part of the manuscript.

- In Figure 2A, the authors claim that there is a stronger effect in old mice, however this is difficult to assess (no statistics is provided for the linear plots).

- In Figure 2E-F, the authors show that in vitro differentiation of a pancreatic cell line exhibit defects in insulin production upon siRNA-MOF, yet such changes do not depend on a transcriptional effect. Although it is clear that this represents a cell autonomous effect for MOF on pancreatic cells, that could partially explain the insulin resistance observed in the mice following HFD, the phenotype is completely independent of the phenotype in fat cells (which is the main topic they follow in the next Figures), and as shown, it is unclear how lack of MOF causes these defects (on that note, it is possible that decreasing MOF influences the differentiation stage of these cells, in which case the decrease insulin production may be due to the fact that these cells are not insulin-secreting cells, something the authors did not assess). I would suggest the authors to move these results to Supplemental Information.

- In Figure 3, the decrease in body weight upon HFD is striking, and indeed a major phenotype in the manuscript. The data indicating that the phenotype is due to lack of fat storage in adipocytes is as well solid and highly conclusive.

- However, they authors claim that they see clear increase glucose levels following the GTT experiment even after 8 weeks of HFD, yet figure 3h shows no difference at 8 weeks, while Ext. Data

Fig 4f it does seem to show a difference. I could not understand what are the differences between these experiments, and why the discrepancy.

- Related to these experiments, the authors included three videos that claim to purport no changes in behavior, feeding or motility, but increase urination in MOF hets. In truth, the videos are short, few seconds videos that are impossible to evaluate (no quantification is provided), and as shown, do not contribute to the study.

- In Figure 4, the authors provide another full omics experiment comparing genome-wide RNA seq data with ChIP-seq data, indicating that thousands of genes (<80% of downregulated genes) are direct targets of MOF. Yet, what the relevance of all these targets in the phenotype remains unclear, particularly when the authors claim that the phenotype is primarily due to MOF effects specifically on PPAR-g, which in turn modulates GLUT4 levels. The authors may consider moving this data to Suppl. Information.

- It is notable that the authors do not provide evidence that upon exposure to HF or glucose, MOF moves to the promoter of PPAR-g to induce expression and upregulation of GLUT-4, as predicted by their model. I understand that performing full ChIP-seq on yet another condition is a lot to ask, but at least the authors could just do ChIP specifically on the PPAR-g promoter, which shouldn't be difficult.

- In Figures 5-6, the authors take advantage of an in-vitro model of fat differentiation to prove that MOF deficiency decreases PPAR-g levels, in turn decreasing GLUT-4, explaining the decrease in glucose uptake and fat accumulation in droplets. Furthermore, they rescue the phenotype using an inhibitor of SIRT1, overexpression of GLUT4 or an activator of PPAR-g, all experiments solidly supporting their model. As I mentioned above, given the importance of this model in the study, the authors should have performed metabolomics in this system (as opposed to the Hela cells) to confirm that they seem similar changes to the ones they see in the mice (of note, in the mouse data, they did not include fat tissue, which is surprising).

Reviewer #2 (Remarks to the Author):

The manuscript by Rodrigues et al. aims to better understand the direct role of the histone acetyltransferase MOF in metabolic dysfunction and predisposition to type 2 diabetes. Using a Mof+/- mouse model, they identify a number of metabolic pathways that are markedly affected, including glucose homeostasis. These mice have altered glucose sensing and are resistant to high fat diet-induced weight gain, which was due in part to defective neutral lipid storage in visceral WAT. To gain more mechanistic insight, they use Mof+/- MEFs and find a relationship between PPARg, Glut4, and impaired glucose uptake and lipid storage. The metabolic profiling of these mice and potential link to human disease is very exciting. The mechanistic studies fall a bit short, but overall the study is intriguing, and only a handful of issues need to be addressed.

1. My main concern is the conclusion that Mof+/- mice have a predisposition to T2D. I think this language needs to be toned down and some caveats of the study and how translatable the results are to human disease discussed.

2. Through the manuscript, there is a lack of H4K16ac quantification. Western blots, or IHC for mouse tissues, should be performed for all experiments. This is especially true in the MEF studies when using MOF inhibitors and Sirt1 inhibitors.

3. Along these lines, ChIP-qPCR using H4K16ac at the PPARg promoter would provide additional rigor to the conclusion that Mof directly regulates PPARg gene expression through H4K16 acetylation. This ideally should be done using the Mof inhibitor and Sirt1 inhibitor.

4. The mechanism of Mof => PPARg => glut4 => glucose uptake/lipid storage is not fully delineated by the experiments performed. To truly show this is specifically through Glut4, a PPARg agonist should

be used in Mof+/-, Glut4 knockdown cells. If the hypothesis is correct, this should not be able to rescue the defect.

Reviewer #3 (Remarks to the Author):

In this manuscript, Akhtar and colleagues performed a comprehensive analysis of the function of acetyltransferase MOF in central carbon metabolism and diet-induced obesity. The data are solid and well presented. However, I have one major concern. Besides acting as a histone acetyltransferase, mammalian MOF has been shown to acetylate the lysine residues in a number of non-histone proteins, including the subunits of multiprotein complexes (e.g., Msl3 and TIP5) and crucial transcription factors (e.g., p53 and Nrf2) [Nat. Cell Biol., 2009, 11, 1010-1016; Mol. Cell, 2006, 24, 841-851]. In fact, a paper published just a few days ago showed that MOF acetylates ER α , which inhibits its ubiquitination and stabilizes it in HCC [Cancer Sci, 2021, DOI: 10.1111/cas.14836]. Since MOF possesses multiple acetylation functions, the authors should address whether these non-histone protein targets also play a role in the phenotypes observed in the Mof+/- mice. To demonstrate this more directly, the authors should construct histone H4K16R and H4K16Q mutations in WT mice or at least in adipocytes as controls to further demonstrate the direct link between histone H4 lysine 16 acetylation and central carbon metabolism or obesity.

Reviewer #4 (Remarks to the Author):

This study from Rodrigues et al. is an in-depth characterization of the effects of MOF-mediated histone acetylation changes in vivo. The work spans a wide range of experiments and suggests a link between MOF down regulation, PPAR γ and GLUT4 which impact glucose uptake and lipid storage. A few comments with respect to work should be considered though the work is quite encompassing.

(1) P.6, ln 151 - I appreciate the investigation of haploinsufficient (MOF+/-) mice as a titrated dose for interrogating the role of MOF in vivo. The extensive metabolomics data in Figure 1 support a role for this perturbation in terms of supporting a change in vivo central carbon metabolism. I am however confused by the comparison to siRNA-mediated depletion of MOF in galactose-adapted HeLa cells. What is the role of this parallel experiment? Perhaps it is to establish early time point changes versus late, but it seems confusing. One would expect these cells to be compared to a tissue of interest in the whole animal experiment, though this is hard to do.

(2) P.9, ln 232 - The authors summarizes stating that the data indicate that Mof reduction affects glucose sensing rather than production. Given that their model is that of human PANC-1 cells differentiated it makes it challenging to separate glucose utilization from sensing. Later in the study they suggest the mechanism is via PPAR γ and GLUT4 is downstream, therefore the mechanism is GLUT4 expression levels? How does this make sense in the context of the HeLa experiments earlier given that the same effect is seen, regulation of central carbon metabolism though they are supplemented with insulin etc.? This progression is confusing and should be flushed out.

(3) P.10, ln 248 - The strongest correlation in the setting of Mof+/- and humans appears to be with amino acid metabolism. This seems like an odd statement to then follow with "Collectively these data identify defective glycolysis and impaired insulin secretion as the major pathological consequences..."

(4) P.18, ln 465 - Measurements of ECAR do not necessarily imply that glucose consumption has changed. The standard in the field would be isotope tracing, but even ³H-glucose is sufficient for uptake. Figure 5 also include 2-NBDG measurements though these are not discussed in the text. One would argue those are better read outs for glucose uptake. Additionally, the authors should include OCR measurements as well to characterize the effect on oxygen consumption (this could be in the extended data).

(5) P.20, ln 502 - GLUT4 protein level is established as the reason for reduced storage of lipids, as

opposed to translocation. The wording of this sentence should be revised as it is confusing. The authors do not probe whether GLUT4 is post-translationally modified. Furthermore, the appropriate experiment would be to rescue in vitro by overexpressing GLUT4 or any other GLUT transporter.

(6) P.20, ln 513 – At this point the authors turn to PPAR γ as the master regulator and this is supported by MOF being bound to the promoter, connecting the dots to GLUT4. Again the key experiment would be to overexpress in the context of MOF knockout adipocytes and rescue the phenotype.

Manuscript NCOMMS-20-50253-T
Point-by-point response to the Reviewers

Summary of major experiments added upon revision

We would like to thank the reviewers for their constructive criticism and suggestions which have enabled us to improve the text and provide experimental data to further strengthen our conclusions through the addition of 18 new panels (Fig 5f-g; Fig 6c-e, 6i, 7d-h, Extended Data Fig. 1b, Extended Data Fig. 2a-c, Extended Data Fig 6b, Extended Data Fig 6j, Extended Data Fig. 7e). In addition to the detailed point-by-point responses for each reviewer, below we have compiled the 4 main comments raised by the reviewers and how we have addressed them in the revised manuscript.

1) Quantification of H4K16ac levels in additional tissues and *in vitro* experiments

We have now provided a comprehensive comparison of several tissues in wild type and *Mof*^{+/-} animals as well as after distinct treatments *in vitro*. The additional data not only confirmed an overall decrease in H4K16ac levels in the *Mof*^{+/-} mutant animals but also shed light onto how dynamic H4K16ac levels are depending a tissue's energetic demand in wild type conditions. Moreover, we further validated the usage of MG149 as a MOF inhibitor, by decreasing the levels of H4K16ac and Ex-527 in promoting H4K16ac levels.

2) Metabolic characterization of MOF function in adipocytes

We have conducted FIA-MS analysis of polar and apolar metabolites in visceral adipocytes from wild type and *Mof* heterozygous mice. In summary we found decreased carbon and amino acid metabolism.

3) Further strengthening of the proposed transcriptional network

Upon revision, we now provide experiments which confirm the relationship and hierarchies between MOF and members of the *Glut4* siRNA treatment in the presence or absence of the PPAR-γ agonist TZD, ChIP-qPCR analyses of MOF and H4K16ac at the *Pparg* promoter in control and *Mof*-iKO models in the presence or absence of glucose/insulin treatment.

4) Statement on correlation of mouse metabolic phenotype with T2D in humans was toned down

We have revised the text to tone down the correlation of our phenotype to human pathologies.

Reviewer #1 (Remarks to the Author):

In this manuscript, Akhtar and colleagues provide a comprehensive metabolic characterization in MOF (the H4K16 Acetyl transferase) +/- mice, indicating that organismal metabolism is clearly affected by altered levels of MOF. They provide further evidence to indicate cell autonomous effects in the pancreas that affect insulin secretion, and in parallel defined MOF as a direct modulator of PPAR-g in fat cells, in turn influencing glucose uptake and lipid droplets formation in these cells. In vivo, such roles protected MOF +/- mice from high fat diet. Overall, the study includes an enormous amount of data from in vivo and in vitro metabolomics, proteomics, RNA sequencing, ChIP-seq and metabolism studies that has not been performed in the past, and as **such it represents useful information to the field**. Yet, the manuscript reads as a collection of different findings, with the authors moving from one to the next without much rationale, and in some cases with findings that are not fully explained by the authors' model.

We appreciate the reviewer for acknowledging the amount of data provided in the manuscript. In the revised manuscript we streamlined the major findings and provided further explanation.

Specific comments

- One major concern relates to the fact that the authors start the paper with a full metabolomics characterization of the different tissues in the MOF hets, that the authors claim they appear much earlier than changes in the proteome. This data suggest that those major metabolic changes are NOT related to transcriptional/translational effects, yet they spend the rest of the manuscript on characterizing direct transcriptional targets of MOF, like PPAR-g. As such, it is unclear why these general metabolic changes occur. The authors may want to move that data to the Suppl. Figures, or else try to explain the molecular reasons for such changes (one possibility: the authors observed major increase in macrophages even in normal diet, suggesting a basal increase in organismal inflammation, which can easily explain the metabolic changes). I do appreciate following such line of studies is well beyond the scope of the study, and hence my suggestion of moving this data to the Suppl. Figures.

We understand the reviewer's concern and agree that the isolated system of cells is probably not closely reflecting the chain of events occurring on the organismal level and as such can lead to misinterpretation of the data. To avoid such unintended confusion, we agree with the reviewer to remove the HeLa dataset.

We agree that inflammation is one of the contributing factors shaping the metabolic profiles of *Mof*^{+/-} mouse tissues. Nonetheless it is important to point out that our data suggests that the lipid storage phenotype is autonomous to adipocytes and therefore independent of the major systemic metabolic issues observed in the *Mof* heterozygous animals. Moreover, we observed impaired central carbon and amino acid metabolism in *Mof*^{+/-} SD WAT (revised Extended Data Fig. 2). However, there was no significant difference in fat storage under these conditions, but instead we observed increased inflammation (Fig. 2a and Extended Data Fig. 4h, Extended Data Fig. 5d). Conversely, we demonstrated that the adipocyte-intrinsic phenotype of impaired lipid storage and by consequence impaired weight gain relies on the intricate transcription network consisting of *Mof*, *Pparg* and *Glut4*. This data suggests that metabolic changes might set the stage for a more pathogenic milieu, which is aggravated under HFD due to the defective *Glut4* transcription network which leads to impaired glucose regulation and lipid storage. It would be indeed fascinating to dissect every single mechanism and the combination of autonomous and paracrine effects in *Mof* heterozygous animals, however this is beyond the scope of this manuscript. Thus, we decided to focus on the most prominent phenotype of impaired fat gain in *Mof* heterozygous animals which is mainly driven by an intricate transcription network.

- In Figure 1D, the authors claim that changes were observed in specific tissues but not in others, however they analyzed much more of the positive tissues (spleen (n = 380), brain (n = 281), heart (n = 209) and serum 117 (n = 293) showed significant alterations, while liver (n = 22), SKM (n = 36) and kidney (n = 4) exhibit no changes. Can the authors explain why the significant difference in tissues analyzed? Such a difference could easily affect the results.

Extended figure 2A: the authors used HeLa cells KD for MOF (adapted on galactose) as supporting evidence for their in vivo metabolomics. These are transformed cells that do not reflect, at all, any relevant tissue, and it is unclear why they included these cells in the manuscript. It will be much more relevant to perform metabolomics in the in vitro differentiated fat cells that they use extensively in the second part of the manuscript.

We apologize if the nomenclature was not clear in the figure. The *n* represents the number of deregulated metabolites found in the tissue. The number of analyzed tissues (sample size) is exactly the same for all the organs (i.e: *Mof*^{+/+} *n*=4 and *Mof*^{+/-} *n*=3). Therefore, the difference in significant deregulated metabolites is rather a biological feature rather than an analytical artefact.

In order to avoid further confusion, we decided to remove the HeLa dataset from our manuscript and replace it with a new FIA-MS analysis obtained from primary using a polar and apolar approach (revised Extended Data Fig. 2). This analysis revealed that adipocytes already at steady state have decreased carbohydrate metabolism, linoleic acid metabolism and amino acid/energy metabolism, while showing an increase of glycosaminoglycan degradation. This suggests that even prior to an HFD challenge, the metabolism of adipocytes from *Mof* heterozygous animals is significantly altered.

- In Figure 2A, the authors claim that there is a stronger effect in old mice, however this is difficult to assess (no statistics is provided for the linear plots).

We apologize if the statistical value was not clearly stated. The difference between young and old mice is indeed significant. In the revised figure, the *p*-value (*p*=0.05) is provided underneath the aging annotation in a bigger font. Moreover, the stronger effect in old mice is also based on insulin response to glucose stimulus (Fig. 2d, 2e), increased glycated hemoglobin (Fig. 3g) and increased free water content (Fig. 3h).

- In Figure 2E-F, the authors show that in vitro differentiation of a pancreatic cell line exhibit defects in insulin production upon siRNA-MOF, yet such changes do not depend on a transcriptional effect. Although it is clear that this represents a cell autonomous effect for MOF on pancreatic cells, that could partially explain the insulin resistance observed in the mice following HFD, the phenotype is completely independent of the phenotype in fat cells (which is the main topic they follow in the next Figures), and as shown, it is unclear how lack of MOF causes these defects (on that note, it is possible that decreasing MOF influences the differentiation stage of these cells, in which case the decrease insulin production may be due to the fact that these cells are not insulin-secreting cells, something the authors did not assess). I would suggest the authors to move these results to Supplemental Information.

We thank the reviewer for their suggestion and agree that it is a phenotype independent of the visceral adipocyte main phenotype in HFD. Therefore, we moved these findings to the supplementary section (Revised Extended Data Fig. 3a). Nonetheless, the knockdown was performed after differentiation. Hence, the cells were insulin-secreting before the knockdown was applied.

- In Figure 3, the decrease in body weight upon HFD is striking, and indeed a major phenotype in the manuscript. The data indicating that the phenotype is due to lack of fat storage in adipocytes is as well solid and highly conclusive.

We highly appreciate that the reviewer considers our results solid and conclusive.

- However, they authors claim that they see clear increase glucose levels following the GTT experiment even after 8 weeks of HFD, yet figure 3h shows no difference at 8 weeks, while Ext. Data Fig 4f it does seem to show a difference. I could not understand what are the differences between these experiments, and why the discrepancy.

We apologize that the results were not represented in a clearer manner. The differences between Fig. 3h and former Extended Data Fig. 4f (now revised Extended Data Fig. 4d-f) is solely the gender of the animals. Fig. 3 shows the results in male animals, while revised Extended Data Fig. 4 shows the results obtained in females. Our data demonstrate that although the overall phenotype is gender independent, the kinetics of the GTT are slightly different. We believe that this might occur mainly due to hormonal differences.

- Related to these experiments, the authors included three videos that claim to purport no changes in behavior, feeding or motility, but increase urination in MOF hets. In truth, the videos are short, few seconds videos that are impossible to evaluate (no quantification is provided), and as shown, do not contribute to the study.

We apologize that the videos were not meaningful, in the revised manuscript we removed them from the supplementary information.

- In Figure 4, the authors provide another full omics experiment comparing genome-wide RNA seq data with ChIP-seq data, indicating that thousands of genes (<80% of downregulated genes) are direct targets of MOF. Yet, what the relevance of all these targets in the phenotype remains unclear, particularly when the authors claim that the phenotype is primarily due to MOF effects specifically on PPAR-g, which in turn modulates GLUT4 levels. The authors may consider moving this data to Suppl. Information.

We kindly disagree with the reviewer's point of view on moving the ChIP-seq dataset to the supplementary section. In Fig. 4 we demonstrate that the pathways associated with MOF bound regions highly correlate with lipid storage. This information was pivotal to prompt us into the direction of interrogating the regulation of fat storage via PPAR- γ , hence the basis for the further evaluation of the aforementioned pathway. We believe that the ChIP-seq profile in adipocytes is relevant to the study and should be presented as the main figure.

- It is notable that the authors do not provide evidence that upon exposure to HF or glucose, MOF moves to the promoter of PPAR-g to induce expression and upregulation of GLUT-4, as predicted by their model. I understand that performing full ChIP-seq on yet another condition is a lot to ask, but at least the authors could just do ChIP specifically on the PPAR-g promoter, which shouldn't be difficult.

We thank the reviewer for their suggestion and we now added the qPCR of MOF and H4K16ac ChIP at the *Pparg* promoter to the revised manuscript (revised Fig. 6i). This experiment revealed increased signal for both MOF and H4K16ac IPs at *Pparg* promoter upon glucose and insulin treatment in iAdipocytes. In line with the narrative of our manuscript, this fails to occur in the *Mof* iKO adipocytes. This additional data supports the notion that MOF-mediated regulation of *Pparg* expression is triggered by insulin/glucose stimulation.

- In Figures 5-6, the authors take advantage of an in-vitro model of fat differentiation to prove that MOF deficiency decreases PPAR-g levels, in turn decreasing GLUT-4, explaining the decrease in glucose uptake and fat accumulation in droplets. Furthermore, they rescue the phenotype using an inhibitor of SIRT1, overexpression of GLUT4 or an activator of PPAR-g, all experiments solidly

supporting their model. As I mentioned above, given the importance of this model in the study, the authors should have performed metabolomics in this system (as opposed to the HeLa cells) to confirm that they seem similar changes to the ones they see in the mice (of note, in the mouse data, they did not include fat tissue, which is surprising).

We appreciate the reviewer's suggestion and removed the HeLa dataset from the manuscript. As discussed above, we added the adipocyte metabolomic analysis as Extended Data Fig. 2 instead.

Reviewer #2 (Remarks to the Author):

The manuscript by Rodrigues et al. aims to better understand the direct role of the histone acetyltransferase MOF in metabolic dysfunction and predisposition to type 2 diabetes. Using a *Mof*^{+/-} mouse model, they identify a number of metabolic pathways that are markedly affected, including glucose homeostasis. These mice have altered glucose sensing and are resistant to high fat diet-induced weight gain, which was due in part to defective neutral lipid storage in visceral WAT. To gain more mechanistic insight, they use *Mof*^{+/-} MEFs and find a relationship between PPAR γ , Glut4, and impaired glucose uptake and lipid storage. The metabolic profiling of these mice and potential link to human disease is very exciting. The mechanistic studies fall a bit short, but overall the study is intriguing, and only a handful of issues need to be addressed.

1. My main concern is the conclusion that *Mof*^{+/-} mice have a predisposition to T2D. I think this language needs to be toned down and some caveats of the study and how translatable the results are to human disease discussed.

In the revised manuscript we have toned down the language with regards to the direct translatability of the results to human disease at this early preclinical stage.

2. Through the manuscript, there is a lack of H4K16ac quantification. Western blots, or IHC for mouse tissues, should be performed for all experiments. This is especially true in the MEF studies when using MOF inhibitors and Sirt1 inhibitors.

We thank the reviewer for pointing out that some validations of H4K16ac levels were indeed missing. In the revised manuscript we show the steady state H4K16ac levels in brain, fat, heart, kidney, liver, skeletal muscle and spleen (Extended Data Fig 1b and c); H4K16ac levels in iAdipocytes after 4OHT in vitro treatment (Fig. 6c, Extended Data Fig. 6f-g), H4K16ac levels in control and *Mof*-iKO iAdipocytes at steady state, upon glucose/insulin treatment, following ectopic expression of *Glut4* with or without glucose/insulin treatment, following ectopic expression of *Mof* and *Mof* E350Q with or without glucose/insulin treatment (Fig. 6d-e); after TZD with or without glucose/insulin treatment (Fig. 7d); upon MG149 (Extended Data. 7b-c) and Ex-527 treatment (Extended Data. 7e). A selective increase of H4K16ac after Ex-527 treatment has also previously been described in PMID: 32671208, PMID: 23863932 and PMID: 29308302.

3. Along these lines, ChIP-qPCR using H4K16ac at the PPAR γ promoter would provide additional rigor to the conclusion that *Mof* directly regulates PPAR γ gene expression through H4K16 acetylation. This ideally should be done using the *Mof* inhibitor and Sirt1 inhibitor.

We appreciate this suggestion and have now added ChIP-qPCR data of H4K16ac and MOF at the *Pparg* promoter at steady state and upon glucose treatment (Fig. 6i). We used the *Mof*-iKO system instead of the MOF inhibitor and glucose/insulin to decrease H4K16ac levels.

4. The mechanism of *Mof* => PPAR γ => glut4 => glucose uptake/lipid storage is not fully delineated by the experiments performed. To truly show this is specifically through Glut4, a PPAR γ agonist should be used in *Mof*^{+/-}, Glut4 knockdown cells. If the hypothesis is correct, this should not be able to rescue the defect.

Consistent with the reviewer's hypothesis, treatment with the PPAR- γ agonist TZD fails to rescue lipid storage in *Mof*-iKO cells treated with *Glut4* siRNAs (new data in Fig. 7e-h). In order to further validate our proposed mechanism, we have added the ectopic *Glut4* expression in control and *Mof*-iKO

iAdipocytes (Fig. 6d). Lipid staining revealed that *Glut4* ectopic expression rescues lipid storage in *Mof*-iKO adipocytes. Given that the regulation of *Glut4* expression is the final point of the proposed transcriptional network, we did not observe a rescue of H4K16ac levels in *Mof*-iKO upon ectopic *Glut4* and *Mof* catalytic mutant expression (Fig. 6d-e). In addition, we performed siRNA-mediated knockdown of *Glut4* using three different siRNAs in control and *Mof*-iKO iAdipocytes (Fig. 7e-h). *Glut4* knockdown is sufficient to impair lipid storage in control cells with no further effect on lipid storage in *Mof*-iKO cells. In summary, this series of experiments firmly support our model that i) the loss of *Glut4* expression is responsible for the lipid storage defects in *Mof*-iKO cells and that ii) MOF regulates *Glut4* expression by modulating transcription of its up-stream effector PPAR- γ .

Reviewer #3 (Remarks to the Author):

In this manuscript, Akhtar and colleagues performed a comprehensive analysis of the function of acetyltransferase MOF in central carbon metabolism and diet-induced obesity. The data are solid and well presented. However, I have one major concern. Besides acting as a histone acetyltransferase, mammalian MOF has been shown to acetylate the lysine residues in a number of non-histone proteins, including the subunits of multiprotein complexes (e.g., Msi3 and TIP5) and crucial transcription factors (e.g., p53 and Nrf2) [Nat. Cell Biol., 2009, 11, 1010-1016; Mol. Cell, 2006, 24, 841-851]. In fact, a paper published just a few days ago showed that MOF acetylates ER, which inhibits its ubiquitination and stabilizes it in HCC [Cancer Sci, 2021, DOI: 10.1111/cas.14836]. Since MOF possesses multiple acetylation functions, the authors should address whether these non-histone protein targets also play a role in the phenotypes observed in the *Mof*^{+/-} mice. To demonstrate this more directly, the authors should construct histone H4K16R and H4K16Q mutations in WT mice or at least in adipocytes as controls to further demonstrate the direct link between histone H4 lysine 16 acetylation and central carbon metabolism or obesity.

We thank the reviewer for considering our data solid and well presented. We agree with the reviewer that the non-histone targets of MOF might provide new insights into the metabolic issues observed in the animals. Nonetheless, in this study we opted to focus on the adipocyte defect in storing lipids, a phenotype that relies on an intricate transcription network in which MOF regulates *Glut4* expression and glucose uptake (Fig. 6 and Fig. 7).

Although we appreciate the spirit of the reviewer's experimental suggestion, histone replacement has unfortunately not yet been described in mice and any phenotypes resulting from overexpression of H4K16Q and R mimics would be extremely difficult to interpret against a wildtype background. However, in order to address the reviewer's suggestion to demonstrate the lipid storage phenotype in more histone direct models we have employed wild type and H4K16R mutant *Drosophila melanogaster*. Those animals were fed under low (30g/L) or high (300g/L) glucose food. After two days we collected the animals and stained the abdomen carcass with red oil to evaluate the fat accumulation. These experiments revealed that H4K16R mutants failed to increase their lipid content under high glucose food, therefore validating the essential role of H4K16ac in regulating glucose homeostasis (Figure for Reviewer - Below).

Figure for Reviewer. H4K16R mutant flies failed to increase their lipid content under high glucose food.

Reviewer #4 (Remarks to the Author):

This study from Rodrigues et al. is an in-depth characterization of the effects of MOF-mediated histone acetylation changes in vivo. The work spans a wide range of experiments and suggests a link between MOF down regulation, PPAR γ and GLUT4 which impact glucose uptake and lipid storage. A few comments with respect to work should be considered though **the work is quite encompassing**.

We appreciate that the reviewer considers our study encompassing, we now addressed their concerns.

(1) P.6, In 151 - I appreciate the investigation of haploinsufficient (MOF $^{+/-}$) mice as a titrated dose for interrogating the role of MOF in vivo. The extensive metabolomics data in Figure 1 support a role for this perturbation in terms of supporting a change in vivo central carbon metabolism. I am however confused by the comparison to siRNA-mediated depletion of MOF in galactose-adapted HeLa cells. What is the role of this parallel experiment? Perhaps it is to establish early time point changes versus late, but it seems confusing. One would expect these cells to be compared to a tissue of interest in the whole animal experiment, though this is hard to do.

We thank the reviewer for their appreciation of our metabolomics analysis. Our original motivation for profiling the metabolites deregulated following MOF depletion in HeLa cells was indeed to try to dissect primary or early effects from late and potentially secondary effects. However, we agree that it is difficult to draw parallels between a human cancer cell line and primary mouse tissue. Therefore, we have opted to remove the HeLa dataset. We have instead added metabolic profiling of white adipose tissue to the revised manuscript (Extended Data Fig. 2).

(2) P.9, In 232 – The authors summarize stating that the data indicate that Mof reduction affects glucose sensing rather than production. Given that their model is that of human PANC-1 cells differentiated it makes it challenging to separate glucose utilization from sensing. Later in the study they suggest the mechanism is via PPAR γ and GLUT4 is downstream, therefore the mechanism is GLUT4 expression levels? How does this make sense in the context of the HeLa experiments earlier given that the same effect is seen, regulation of central carbon metabolism though they are supplemented with insulin etc.? This progression is confusing and should be flushed out.

In order to better characterize the glucose sensing defect and dissect the relationship between MOF, PPAR- γ and GLUT4 we have opted to focus on WAT/adipocytes and eliminate the HeLa data. In the previous version of the manuscript we had already shown a significant reduction in *Ppar γ* , *Pgc1 α* , *Mef2c* and *Glut4* in WAT of SD-fed *Mof $^{+/-}$* mice (Fig. 6g-i) and that the *Ppar γ* agonist TZD partially rescued lipid storage defects in *Mof*-depleted cells (Fig. 7), suggesting that MOF acts upstream of PPAR- γ . During revision we solidified this by adding CHIP-qPCR data showing that MOF binds directly to the *Pparg* promoter in both WAT and iAdipocytes (Fig. 6h-i). Therefore, our data supports a model in which the direct regulation of *Pparg* transcription by MOF subsequently has a dramatic effect on *Glut4* transcription and in turn overall GLUT4 levels (see Fig 7i for visual representation, Fig. 2b-e, Fig. 5i-k).

(3) P.10, In 248 – The strongest correlation in the setting of *Mof $^{+/-}$* and humans appears to be with amino acid metabolism. This seems like an odd statement to then follow with “Collectively these data identify defective glucose assimilation and impaired insulin secretion as the major pathological consequences...”

Upon revision, we have adjusted the tone in the revised manuscript. However, we would like to point out that the molecular similarities between *Mof $^{+/-}$* mice and human T2D pathologies are not limited to amino acid deregulation, *Mof $^{+/-}$* animals and human T2D patients also exhibit parallels in their

metabolic profiles (Fig. 2g), gene regulatory networks (Fig. 2f), the increase in glycated hemoglobin (Fig. 3g), the increased free water content (Fig. 3h) and the increased levels of IL17a (Ext. Data. Fig. 3i-m), and unbalance glutamine to glutamate ratio (Fig. 1h), thus suggesting a multifaceted correlation with human T2D. Collectively, these data identify defective glucose assimilation and impaired insulin secretion as one of the major pathological consequences of *Mof* haploinsufficiency.

(4) P.18, In 465 – Measurements of ECAR do not necessarily imply that glucose consumption has changed. The standard in the field would be isotope tracing, but even ³H-glucose is sufficient for uptake. Figure 5 also include 2-NBDG measurements though these are not discussed in the text. One would argue those are better read outs for glucose uptake. Additionally, the authors should include OCR measurements as well to characterize the effect on oxygen consumption (this could be in the extended data).

Thank you for bringing this to our attention, we now explain the 2-NBDG results in Fig. 5j in the text. We have added OCR measurements to the revised manuscript as panel (j) in Extended Data. Fig. 6. No significant differences between *Mof*-iKO and control iAdipocytes were observed under steady state, but a tendency to decrease OCR after combined insulin and glucose treatment.

(5) P.20, In 502 – GLUT4 protein level is established as the reason for reduced storage of lipids, as opposed to translocation. The wording of this sentence should be revised as it is confusing. The authors do not probe whether GLUT4 is post-translationally modified. Furthermore, the appropriate experiment would be to rescue in vitro by overexpressing GLUT4 or any other GLUT transporter.

The successful reversal of lipid storage defects in *Mof*-iKO iAdipocytes via *Glut4* overexpression is shown in Fig. 6d. Moreover, we could rescue lipid storage using a PPAR- γ agonist in the *Mof*-iKO background but not following combined depletion of both *Mof* and *Glut4* (Fig. 7e-h). This finding supports our idea that MOF, and GLUT4 function in the same pathway, with GLUT4 upstream and PPAR- γ downstream of MOF. We have now rewritten the confusing sentence.

(6) P.20, In 513 – At this point the authors turn to PPAR γ as the master regulator and this is supported by MOF being bound to the promoter, connecting the dots to GLUT4. Again the key experiment would be to overexpress in the context of MOF knockout adipocytes and rescue the phenotype.

This point was addressed above. We were also able to restore the lipid storage capacity and transcription of *Pcg1a* and *Glut4* in *Mof*-iKO iAdipocytes using TZD, a known PPAR γ agonist (PMID: 16248830) (Fig.7a-c). In parallel, we observed that TZD treatment failed to rescue *Mof*-iKO iAdipocytes which are depleted of *Glut4*. Taken together, our data suggest that MOF orchestrates glucose uptake and subsequent lipid storage by transcriptional regulation of *Glut4* by mediating *Ppar γ* expression.

REVIEWERS' COMMENTS

Reviewer #1 (Remarks to the Author):

The authors did a major effort in responding to the reviewers' concerns, and the manuscript is significantly improved. In particular, the addition of the metabolomics in adipocytes, the new ChIP data on the Pparg promoter, the new H4K16Ac quantifications, and the Glut4 KD/OE rescue experiments, all significantly strengthen the manuscript. I believe it will be of broad interest to the readership of Nat.Comm. I do not have any more concerns.

Reviewer #2 (Remarks to the Author):

The authors have addressed my concerns with many new pieces of data. The mechanistic studies are now more rigorous. No further concerns are noted.

Reviewer #3 (Remarks to the Author):

The authors have answered all of my questions and have provided additional explanations/clarifications where required. I am happy to accept this revised version as is and have no further remarks.

Reviewer #4 (Remarks to the Author):

The authors have addressed all of my previous comments.

Manuscript NCOMMS-20-50253A

Point by point response

Reviewer #1

The authors did a major effort in responding to the reviewers' concerns, and the manuscript is significantly improved. In particular, the addition of the metabolomics in adipocytes, the new ChIP data on the Pparg promoter, the new H4K16Ac quantifications, and the Glut4 KD/OE rescue experiments, all significantly strengthen the manuscript. I believe it will be of broad interest to the readership of Nat.Comm. I do not have any more concerns.

We thank the reviewer for the supportive comments and are delighted that the reviewer was happy with the revised manuscript.

Reviewer #2

The authors have addressed my concerns with many new pieces of data. The mechanistic studies are now more rigorous. No further concerns are noted.

We thank the reviewer for the supportive comments and are delighted that the reviewer was happy with the revised manuscript.

Reviewer #3

The authors have answered all of my questions and have provided additional explanations/clarifications where required. I am happy to accept this revised version as is and have no further remarks.

We thank the reviewer for the supportive comments and are delighted that the reviewer was happy with the revised manuscript.

Reviewer #4

The authors have addressed all of my previous comments.

We thank the reviewer for the supportive comments and are delighted that the reviewer was happy with the revised manuscript.